# Exploitation of the fibrinolytic system by B-cell acute lymphoblastic leukemia and its therapeutic targeting

Valentina R. Minciacchi [1], Jimena Bravo[2], Christina Karantanou[3], Raquel S. Pereira [4], Costanza Zanetti[5], Rahul Kumar[2], Nathalie Thomasberger[6], Pablo Llavona[7], Theresa Krack[2], Katrin Bankov[8], Melanie Meister[9], Sylvia Hartmann [10], Véronique Maguer-Satta [11], Sylvain Lefort [11], Mateusz Putyrski[12], Andreas Ernst[13], Brian J. P. Huntly [14], Eshwar Meduri[14], Wolfram Ruf [1,15] & Daniela S. Krause [2,16,17,18] ✉

Fibrinolysis influences the mobilization of hematopoietic stem cells from their bone marrow microenvironment (BMM). Here we show that activation of plasmin, a key fibrinolytic agent, by annexin A2 (ANXA2) distinctly impacts progression of BCR-ABL1⁺ B-cell acute lymphoblastic leukemia (B-ALL) via modulation of the extracellular matrix (ECM) in the BMM. The dense ECM in a BMM with decreased plasmin activity entraps insulin-like growth factor (IGF) 1 and reduces mTORC2-dependent signaling and proliferation of B-ALL cells. Conversely, B-ALL conditions the BMM to induce hepatic generation of plasminogen, the plasmin precursor. Treatment with ε-aminocaproic acid (EACA), which inhibits plasmin activation, reduces tumor burden and prolongs survival, including in xenogeneic models via increased fibronectin in the BMM. Human data confirm that IGF1 and fibronectin staining in trephine biopsies are correlated. Our studies suggest that fibrinolysis-mediated ECM remodeling and subsequent growth factor release influence B-ALL progression and inhibition of this process by EACA may be beneficial as adjunct therapy.

The type of leukemia and even the type of oncogene[1] determine the reciprocal interactions of leukemia cells with cellular and acellular components of the bone marrow (BM) microenvironment (BMM) in a highly specific way. Such interactions lead to establishment of a distinct, protective and supportive environment for leukemic stem cells (LSC) and are major contributors to inefficient disease eradication by many treatments[2], raising the need for novel therapies.

[1]Center for Thrombosis and Hemostasis (CTH), Johannes Gutenberg University Medical Center, 55131 Mainz, Germany. [2]Institute of Transfusion Medicine – Transfusion Center, Johannes Gutenberg University Medical Center, 55131 Mainz, Germany. [3]Department of Vascular Dysfunction - Medical Faculty Mannheim of Heidelberg University, Mannheim, Germany. [4]Institute for Experimental Pediatric Hematology and Oncology, Goethe-University Frankfurt, Frankfurt am Main, Germany. [5]Division of mRNA Cancer Immunotherapy, Helmholtz Institute for Translational Oncology Mainz, Mainz, Germany. [6]BioNTech SE, Mainz, Germany. [7]The Institute of Molecular Biology, Mainz, Germany. [8]Department of Pediatrics (Hematology/Oncology), Charité-Universitätsmedizin, Berlin, Germany. [9]AbbVie, Wiesbaden, Germany. [10]Department of Pathology, Goethe University, Frankfurt am Main, Germany. [11]CRCL, Inserm U1052-CNRS UMR5286, Centre Léon Bérard, Lyon, France. [12]Fraunhofer Institute for Molecular Biology and Applied Ecology IME, Project Group Translational Medicine & Pharmacology TMP, Frankfurt am Main, Germany. [13]Pharmazentrum/ZAFES Frankfurt, Faculty of Medicine, Goethe-University Frankfurt, Frankfurt am Main, Germany. [14]Department of Haematology, University of Cambridge, Cambridge, UK. [15]Department of Immunology and Microbiology, Scripps Research, La Jolla, CA, USA. [16]German Cancer Research Center (DKFZ), Heidelberg, Germany. [17]German Cancer Consortium (DKTK), Heidelberg, Germany. [18]Research Center for Immunotherapy (FZI), University Medical Center, University of Mainz, Mainz, Germany. ✉e-mail: krauseda@uni-mainz.de

Accordingly, we found that Col2.3 kb GFP+ mesenchymal (osteoblastic) cells[3] from mice with BCR-ABL1+ chronic myeloid leukemia (CML) versus MLL-AF9+ acute myeloid leukemia (AML) showed higher expression of the annexins (*Anxa*) A2 and A5[4], which belong to the family of $Ca^{2+}$-dependent membrane proteins[5]. This suggested possible regulation of leukemia progression by these BMM factors. Indeed, we recently showed a deficiency of ANXA5 to be involved in the creation of an inflammatory BMM, accelerating AML progression[4].

ANXA2, on the other hand, facilitates the homing of multiple myeloma cells to the BM[6]. In leukemia, a role for ANXA2 has only been reported on tumor cells[7]. Cell surface ANXA2 forms a heterotetramer with S100A10, a member of the S100 family of proteins, facilitating the conversion of plasminogen (PLG) to plasmin by tissue plasminogen activator (tPA) and, thereby, acting as activator of fibrinolysis[8]. In solid cancers, activation of plasmin, a protease essential for fibrinolysis, promotes disease progression and metastasis via degradation of extracellular matrix (ECM) proteins[9], activation of matrix metalloproteinases (MMPs)[10] and release of growth factors (GFs) from the ECM[11]. Therefore, we hypothesized that the fibrinolytic system, previously implicated in the mobilization of hematopoietic stem and progenitor cells[12], including ANXA2, have an impact on leukemia progression, possibly via involvement of the natural anticoagulant pathway and ECM remodeling.

Focusing on another BCR-ABL1+ disease, B-cell acute lymphoblastic leukemia (B-ALL), a target of several novel therapies[13], in which the BMM is known to play an important role[14,15], we unravel here how the fibrinolytic system and its degradation of the ECM in the BMM specifically promotes B-ALL progression. In turn, conditioning of hepatocytes by a leukemic environment leads to the secretion of plasminogen, thereby, perpetuating this leukemia-propagating circuit. We demonstrate the beneficial effect of combination treatment with cytarabine plus the inhibitor of plasmin activation, ε-aminocaproic acid (EACA), in B-ALL, in preclinical models and provide human data supporting the prognostic and therapeutic relevance of interfering with fibrinolysis in human B-ALL.

## Results

### ANXA2-deficiency in the BMM leads to survival extension in BCR-ABL1+ leukemia

Based on our evidence that *Anxa2* was differentially regulated in Col2.3+ cells in AML versus CML (Supplementary Fig. 1A, B)[4], we showed that *Anxa2* expression was higher in Sca1+ CD73+ mesenchymal stromal cells (MSC) from an AML versus a CML BMM (Supplementary Fig. 1C, D, Tables 1, 2). In healthy mice levels of ANXA2 were variable (Supplementary Fig. 1E). These data suggest that gene expression of mesenchymal cells in the murine BMM is differentially impacted by MLL-AF9+ AML versus CML cells.

To identify the contribution of BMM-associated ANXA2 to leukemia progression, as suggested by our microarray data, we transplanted WT leukemia-initiating cells into WT or ANXA2-deficient recipient mice using the retroviral transduction/transplantation models of BCR-ABL1-induced CML or B-ALL or MLL-AF9-induced AML[1]. Hereby, the bone marrow of WT donors (pretreated with 5-fluorouracil in the case of CML and AML, but not B-ALL) is transduced with retroviri expressing BCR-ABL1 (for CML and B-ALL) or MLL-AF9 (for AML) and transplanted into WT or ANXA2-KO recipients. Healthy ANXA2 KO mice, which are viable and fertile[16], had no abnormalities in relevant basic hematological parameters (Supplementary Fig. 2) and hematopoietic stem and progenitor cells (Supplementary Figs. 3, 4) compared to wildtype (WT) mice. In follow-up of our microarray, we first initiated CML experiments. This revealed that induction of CML led to a significant survival extension in ANXA2-deficient compared to WT recipient mice (Supplementary Fig. 5A, B), but homing of CML-initiating cells to WT or ANXA2-deficient environments did not differ (Supplementary Fig. 5C). Myeloid colony formation by BM cells from ANXA2-deficient recipients

with CML was reduced (Supplementary Fig. 5D), and survival of secondary WT recipients of BM, but not spleen cells from ANXA2-deficient donors with CML was significantly prolonged (Supplementary Fig. 5E). In contrast, no significant differences were observed in survival between WT or ANXA2-deficient recipients with MLL-AF9-induced AML (Supplementary Fig. 5F, G).

Induction of BCR-ABL1+ B-ALL (henceforth termed B-ALL), however, led to a significant reduction of GFP+ (BCR-ABL1)+ BP1+ pre-B cells in peripheral blood (PB) (Fig. 1A), as well as survival extension in ANXA2-deficient compared to WT recipient mice (Fig. 1B). BM cells from ANXA2-deficient mice with B-ALL showed reduced ability to form colonies (Fig. 1C) and mostly failed to induce disease in secondary WT recipients (Supplementary Fig. 5H). GFP (BCR-ABL1)+ BP1+ B-ALL-initiating cells homed less to the BM, but not the spleen of ANXA2 KO mice (Supplementary Fig. 5I). Intrafemoral transplantation of B-ALL-initiating cells partially rescued disease induction in ANXA2 KO recipients (Supplementary Fig. 5J, K). Non-irradiation of recipients prior to transplantation[14] led to more pronounced survival extension in ANXA2 KO mice (Fig. 1D, E). Further, infiltration of the majority of organs with B-ALL cells was reduced in irradiated (Supplementary Fig. 6A–G) ANXA2 KO compared to WT recipients. In contrast, ANXA2 deficiency in B-ALL-initiating cells (LIC) did not lead to changes in survival compared to WT LIC (Supplementary Fig. 6H). Lastly, transplantation of empty vector-transduced BM cells into WT versus ANXA2 KO recipients only led to slightly increased percentages of GFP+ CD11b+ myeloid cells in PB and BM of ANXA2 KO mice, but otherwise showed no differences (Supplementary Fig. 7A–J). In summary, with a focus on B-ALL due to the greater need for improved therapies compared to CML, which in most patients is well controlled by treatment with tyrosine kinase inhibitors[17]—these results suggest that microenvironmental ANXA2 deficiency attenuates induction of BCR-ABL1+ B-ALL, possibly due to inhospitality of the BMM. Survival extension in BCR-ABL1+ B-ALL was partially due to reduced LIC homing to an ANXA2 KO BMM, but incomplete rescue of B-ALL in intrafemorally transplanted ANXA2 KO mice and more pronounced prolongation of B-ALL survival in non-irradiated recipients suggest major contributory roles of an ANXA2-deficient BMM for B-ALL progression.

### Plasminogen activation contributes to ECM remodeling in the leukemic BMM

As ANXA2 supports plasminogen activation, we hypothesized that an ANXA2-deficient BMM impairs homing, engraftment and progression of BCR-ABL1+ B-ALL via an accumulation of ECM proteins. Indeed, levels of the ECM protein fibronectin were increased in the BMM of ANXA2-deficient compared to WT mice with B-ALL (Fig. 2A, Supplementary Fig. 8A). Laminin, another ECM protein in the BM[18], in contrast, did not differ significantly in ANXA2 KO versus WT mice (Supplementary Fig. 8B, C). Fibronectin staining in other tissues, which was largely localized around vessels, was only increased in spleens of mice with BCR-ABL1+ B-ALL (Supplementary Fig. 8D). Fibronectin levels were also increased in the BMM of ANXA2-deficient mice transplanted with empty vector-transduced BM and of non-irradiated ANXA2-deficient recipient mice with B-ALL compared to WT mice (Supplementary Fig. 8E, F). Irradiation did not alter preexisting differences in fibronectin levels between WT and ANXA2 KO mice (Supplementary Fig. 8G).

Next, we focused on MSC as important producers of ECM proteins and components of the B-ALL-niche[19] for in vitro experiments as model system, as primary osteoblastic cells, the object of our earlier microarray studies (Supplementary Fig. 1A), which are derived from MSC, cannot be cultured easily. BMM-derived primary murine MSC[20] indeed express high levels of ANXA2 (Supplementary Fig. 9A–C). Immunophenotyping (Supplementary Fig. 9D) and RNA-sequencing (Supplementary Fig. 10A, B) showed no relevant differences between WT and ANXA2-deficient MSC, including with regards to the percentage of

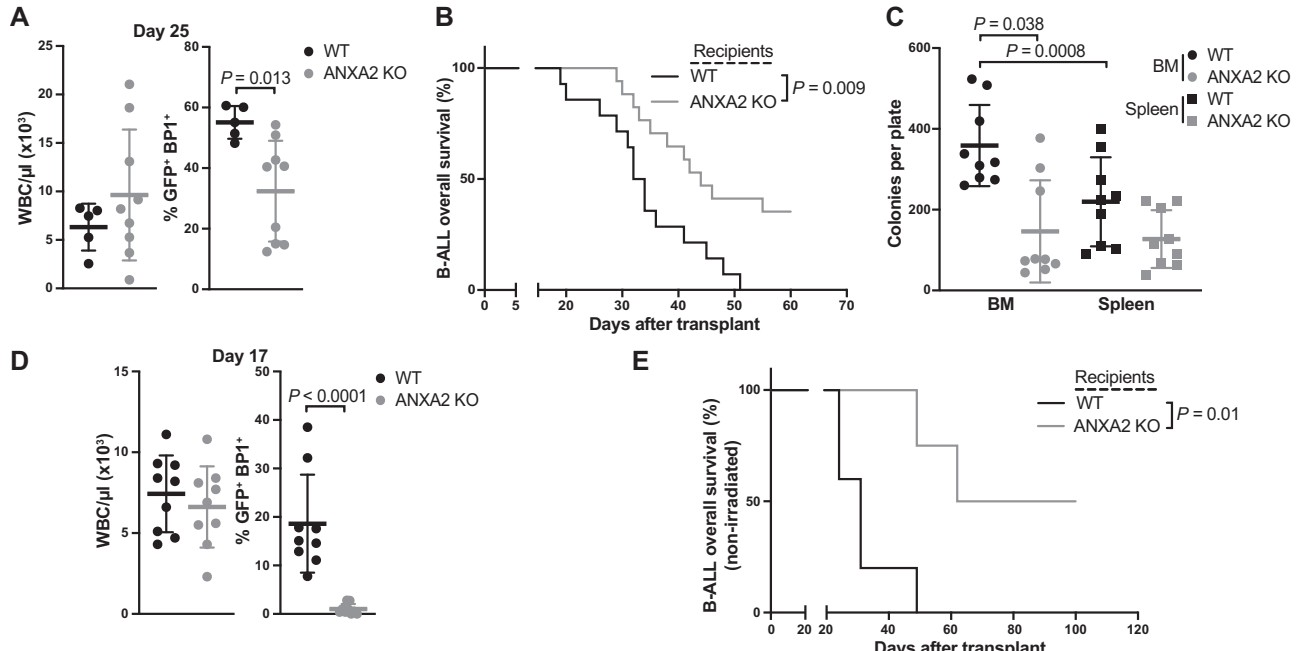

**Fig. 1 | ANXA2-deficiency in the BMM leads to survival extension in BCR-ABL1+ leukemia.** **A** WBC count per μl (left) and percentage of GFP (BCR-ABL1)+ BP1+ cells of all cells (right) in the peripheral blood of WT (black) or ANXA2 KO (gray) recipient mice with BCR-ABL1+ B-ALL ($P = 0.013$, two-tailed $t$ test, WT $n = 5$, ANXA2 KO $n = 10$, mean ± SD). **B** Kaplan–Meier-style survival curve of WT (black) or ANXA2 KO (gray) recipient mice with BCR-ABL1+ B-ALL ($P = 0.009$, Log-rank test, WT $n = 14$, ANXA2 KO $n = 17$). **C** Number of colonies per plate derived from total BM or spleen cells from WT (black) or ANXA2 KO (gray) recipient mice with BCR-ABL1+ B-ALL ($P = 0.038$, $P = 0.0008$, two-way ANOVA, Tukey test, $n = 3$ (3 replicates of 3 individual mice per group), mean ± SD). **D** WBC count per μl (left) and percentage of GFP (BCR-ABL1)+ BP1+ cells (right) of all cells in the peripheral blood of non-irradiated WT (black) or ANXA2 KO (gray) recipient mice with BCR-ABL1+ B-ALL ($P < 0.0001$, two-tailed $t$ test, WT $n = 9$, ANXA2 $n = 9$, mean ± SD). **E** Kaplan–Meier-style survival curve of non-irradiated WT (black) or ANXA2 KO (gray) recipient mice with BCR-ABL1+ B-ALL ($P = 0.01$, Log-rank test, WT $= 5$, ANXA2 $n = 4$). Source data are provided as a Source Data file.

MSC of total live stromal cells (Supplementary Fig. 10C). *Anxa2* expression was also similar between primary murine MSC at baseline and after differentiation into osteoblasts (Supplementary Fig. 10D). A trend towards reduced generation of fibroblast colony forming units (Supplementary Fig. 11A) and a significant decrease of osteoblastic differentiation (Supplementary Fig. 11B) were found in ANXA2 KO compared to WT MSC, whereby adipocyte differentiation did not differ (Supplementary Fig. 11C). The percentage of F4/80+ CD169+ macrophages (Supplementary Fig. 11D), recently shown to play a role in the B-ALL BMM[14,15], and fibroblast morphology (Supplementary Fig. 11E) also did not differ between WT and ANXA2 KO mice.

ANXA2 expression was similar between primary murine MSC, macrophages and fibroblasts (Supplementary Fig. 11F), as well as in murine stroma (MS5), endothelial (H5V) and fibroblastic (3T3) cell lines (Supplementary Fig. 11G), underlining the likely contributory roles of other cell types of the BMM in addition to our chosen model system, the MSCs. Testing a possible ANXA2-mediated contribution to neoangiogenesis[21] to our observed leukemia phenotypes, we found the percentage of CD45- CD31+ EMCN (endomucin)+ endothelial cells to be increased in the BM of ANXA2 KO mice (Supplementary Fig. 11H, I).

We confirmed by co-immunoprecipitation that ANXA2 stabilizes and interacts with S100A10 (Supplementary Fig. 12A). Plasminogen activation assays with WT versus ANXA2-deficient cells confirmed the lack of a functional ANXA2/S100A10 complex and plasminogen activation on ANXA2-deficient MSC, macrophages, fibroblasts (Fig. 2B) and ANXA2-deficient MS5, H5V and 3T3 cells (Supplementary Fig. 12B, C, Table 3). This was accompanied by reduced extracellular laminin and fibronectin associated with WT, but not ANXA2-deficient MSC, whereby fibronectin has previously been described as plasmin substrate[22,23] (Fig. 2C, Supplementary Fig. 12D). Lack of ANXA2 on MSC prevented the invasion of BCR-ABL1+ BA/F3 cells, a frequently used model for BCR-ABL1+ B-ALL cells[19], through a layer of MSC, embedded

in matrigel (Fig. 2D; Supplementary Fig. 12E). However, transient overexpression of ANXA2 in ANXA2-deficient MSC (Supplementary Fig. 12F-G) restored BCR-ABL1+ BA/F3 invasion through MSC embedded in matrigel (Fig. 2D). ANXA2-dependent activation of plasmin also resulted in increased activation of MMP9[24] in WT, but not ANXA2-deficient MSC, specifically after stimulation with tumor necrosis factor (TNF)α, which is known to be secreted by B-ALL cells[19] (Supplementary Fig. 12H).

To confirm the importance of plasminogen activation for B-ALL progression, we transplanted BCR-ABL1-transduced BM cells into WT or tPA-deficient mice. PB leukocytes and GFP+ (BCR-ABL1)+ BP1+ pre-B cells (Supplementary Fig. 12I) were reduced, and survival was significantly prolonged in tPA-deficient compared to WT mice (Fig. 2E). A similar result was obtained in secondary recipients transplanted with BM cells from WT or tPA-deficient mice with B-ALL (Supplementary Fig. 12J, K). Furthermore, BM cells isolated from leukemic tPA-deficient mice failed to form colonies in vitro (Fig. 2F). In line with data from ANXA2 KO mice, immunofluorescence (IF) analysis of bone sections revealed higher fibronectin levels in the BM of tPA-deficient compared to WT mice with BCR-ABL1+ B-ALL (Fig. 2G). Consistently, intrafemoral administration of tPA to WT or ANXA2-deficient mice, in order to bypass ANXA2-dependent plasminogen activation, significantly decreased fibronectin levels in ANXA2 KO, but not in WT mice, in which plasmin is normally activated (Fig. 2H, Supplementary Fig. 12L).

Systemic plasmin administration to mice with B-ALL, which accumulated in the BMM (Supplementary Fig. 13A), however, led to acceleration and a 'rescue' of the disease (Fig. 2I), as well as significant reduction of fibronectin in the BM of ANXA2-deficient mice (Fig. 2J, Supplementary Fig. 13B). In support of specific roles of plasmin for cleaving protein targets[25], recipient mice deficient for protease-activated receptor (PAR)-1, which is activated by plasmin[26], transplanted with B-ALL LIC phenocopied the survival extension observed

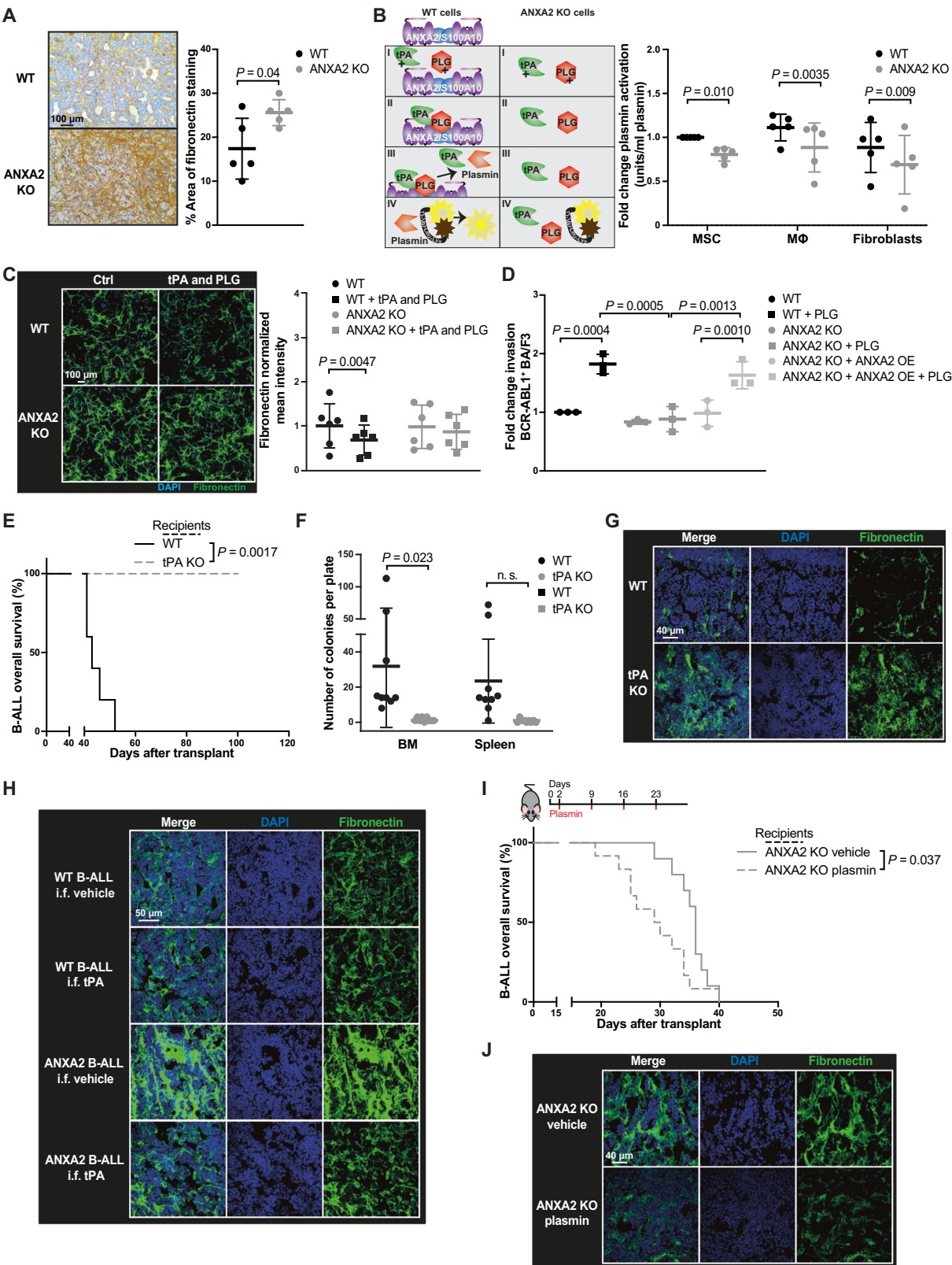

in ANXA2-deficient mice (Supplementary Fig. 13C, D). Consistently, urokinase-type plasminogen activator receptor (uPAR), the receptor for urokinase plasminogen activator (uPA), known to be involved in degradation of the ECM, as well as invasion and metastasis of malignant tumors[27], was higher in the plasma of WT mice with B-ALL than healthy mice (Supplementary Fig. 13E). Thus, ANXA2-mediated plasminogen activation is central to BCR-ABL1⁺ B-ALL development.

## B-ALL promotes hepatic generation of plasminogen

Next, we tested levels of plasminogen, plasmin and tPA in WT or ANXA2 KO mice with B-ALL. Consistent with ANXA2's role in converting plasminogen to plasmin, plasmin, but not plasminogen or tPA, was reduced in the BM or plasma of healthy ANXA2 KO mice or ANXA2 KO mice with B-ALL compared to the respective WT controls (Fig. 3A–C, Supplementary Fig. 14A–C). However, levels of plasmin,

**Fig. 2 | Plasminogen activation contributes to ECM remodeling in the leukemic BMM. A** Immunohistochemistry of fibronectin in bones of WT (black) or ANXA2 KO (gray) recipient mice with BCR-ABL1[+] B-ALL ($P = 0.04$, two-tailed $t$ test, $n = 5$ mice per group, mean ± SD). **B** Active plasmin in the medium of primary WT (black) or ANXA2 KO (gray) mesenchymal stromal cells (MSC), macrophages (MΦ) or fibroblasts ($P = 0.01$, $P = 0.0035$, $P = 0.009$, two-way ANOVA, Sidak test, $n = 5$ biological replicates, mean ± SD), normalized to WT. **C** Immunofluorescence of fibronectin (green) and 4′,6-diamidino-2-phenylindole (DAPI, blue) with or without activation of plasmin in cultures of WT (black) or ANXA2 KO (gray) MSC ($P = 0.0047$, two-way ANOVA, Sidak test, $n = 6$ biological replicates, mean ± SD), normalized to cell number. **D** Invasion of BCR-ABL1[+] BA/F3 cells plated on top of WT (black) or ANXA2 KO MSC (gray) or ANXA2 KO MSC overexpressing (OE) ANXA2 (light gray) ($P = 0.0004$, $P = 0.0005$, $P = 0.0013$, $P = 0.001$, two-way ANOVA, Tukey test $n = 3$ biological replicates, mean ± SD), normalized to WT. **E** Kaplan–Meier-style survival curve of WT (black) versus tPA-knockout (tPA KO; gray) recipient mice with

BCR-ABL1[+] B-ALL ($P = 0.0017$, Log-rank test, WT $n = 5$, tPA KO $n = 5$). **F** Number of colonies per plate derived from total bone marrow or spleen cells from WT (black) or tPA KO (gray) recipient mice with BCR-ABL1[+] B-ALL ($P = 0.023$, two-way ANOVA, Tukey test, $n = 3$ (3 replicates of 3 individual mice per group), mean ± SD). **G** Immunofluorescence of bones, stained for fibronectin (green) and DAPI (blue), from WT or tPA KO recipient mice with BCR-ABL1[+] B-ALL ($n = 3$ mice per group). **H** Immunofluorescence of bones, stained for fibronectin (green) and DAPI (blue), from WT or ANXA2 KO recipient mice with BCR-ABL1[+] B-ALL. Vehicle or tPA was administered by intrafemoral (i.f.) injection ($n = 5$ mice per group). **I** Kaplan–Meier-style survival curve of ANXA2 KO recipient mice with B-ALL treated with vehicle (solid line) or plasmin (dashed line) ($P = 0.037$, Log-rank test, vehicle $n = 10$, plasmin $n = 12$). **J** Immunofluorescence of bones, stained for fibronectin (green) and DAPI (blue), from ANXA2 KO recipient mice with BCR-ABL1[+] B-ALL treated with vehicle or plasmin ($n = 4$ mice per group). Source data are provided as a Source Data file.

plasminogen and tPA were higher in the BM of WT mice with B-ALL compared to healthy mice (Fig. 3D–F), although we cannot exclude that the employed antibody to plasminogen may also be detecting plasmin. D-dimers, representing fibrinogen degradation products as a result of degradation by plasmin, were similar in the plasma of healthy WT versus ANXA2 KO mice (Supplementary Fig. 14D), but increased in the plasma of WT mice with B-ALL compared to healthy controls (Supplementary Fig. 14E).

Plasminogen activator inhibitor (PAI)-1, an inhibitor of tPA, was significantly reduced in WT mice with B-ALL compared to healthy mice (Fig. 3G). The anti-protease α2-macroglobulin was significantly decreased in healthy ANXA2 KO compared to healthy WT mice (Supplementary Fig. 14F), while there was a trend towards decreased levels of α2-macroglobulin in the plasma of WT mice with B-ALL compared to healthy mice (Supplementary Fig. 14G). α2-antiplasmin was unchanged (Supplementary Fig. 14H, I).

Next, we hypothesized, that leukemia cells influence the generation of plasminogen by hepatic cells, either directly or by conditioning of the BMM, which may secrete a mediator of hepatic plasminogen production. Given that interleukin-6 (IL-6) is known to regulate gene expression of plasminogen in hepatocytes[28], and may be secreted by B cells[29] or cells of the BMM, we measured IL-6 levels. We observed that IL-6 was higher in the plasma and liver of WT mice with B-ALL compared to healthy mice (Fig. 3H, I). Consistently, incubation of Huh7 hepatic cells with recombinant IL-6 increased the expression of plasminogen (Fig. 3J).

In a chromatin immunoprecipitation (ChIP) assay using Huh7 cells treated with vehicle or IL-6, we showed that IL-6 significantly increased the binding of cAMP Responsive Element Binding Protein 1 (CREB1), one of the transcription factors downstream of the IL-6 receptor, to the *PLG* regulatory element (Fig. 3K, Supplementary Table 4). In light of our published studies that B-ALL cells produce high amounts of TNFα[19,30] and that TNFα increases IL-6 expression in MSC[4], we speculated that MSC and/or other cell types in the BMM conditioned by B-ALL cells are the likely source of IL-6 leading to hepatic generation of plasminogen. Indeed, treatment of human MSC with TNFα or coculture of human MSC with the BCR-ABL1[+] B-ALL cell line SUPB15, which produces TNFα (Supplementary Fig. 14J), and subsequent culture of Huh7 cells in the conditioned medium of these MSC led to increased levels of plasminogen in the conditioned medium of Huh7 cells (Supplementary Fig. 14K). There was a strong trend towards inhibition of this increase in plasminogen in the conditioned medium of Huh7 cells, if the Huh7 cells, cultured in the conditioned medium from the TNFα-exposed MSC, were simultaneously treated with an antibody to IL-6 (Supplementary Fig. 14L). Taken together, these data suggest that B-ALL-cell derived TNFα conditions MSC and possibly other cell types in the BMM to secrete IL-6, which promotes hepatic generation of plasminogen. In the presence of B-ALL levels of some inhibitors of the plasmin system are reduced.

## Restricted access to growth factors in the BMM leads to reduced mTORC2 activation and proliferation of BCR-ABL1[+] cells

Using ANXA2-deficient MSC as a BMM model of impaired plasminogen activation and ECM accumulation, we evaluated the proliferation of BCR-ABL1[+] BM cells plated in limiting dilution[31] on MSC and showed that leukemia cells were slower at reaching confluence on ANXA2-deficient versus WT MSC (Supplementary Fig. 15A). Immunoblot analysis of sorted primary murine B-ALL cells for downstream targets of the BCR-ABL1 oncoprotein revealed that AKT phosphorylation was significantly decreased in B-ALL cells from the BM (Fig. 4A), but not the spleen (Supplementary Fig. 15B), of ANXA2-deficient compared to WT mice. BCR-ABL1 mediates AKT phosphorylation on the T308 residue (pAKT[T308])[32], while phosphorylation of the S473 residue (pAkt[S473]) is mainly regulated by mTOR complex 2 (C2)[33,34]. mTORC2 lies downstream of receptors for growth factors such as insulin-like growth factor 1 (IGF1)[35,36]. Indeed, IF analysis showed increased colocalization of fibronectin and IGF1 in the BM of mice lacking ANXA2 compared to WT mice, both in the healthy condition (Supplementary Fig. 15C–F) and in B-ALL (Fig. 4B, Supplementary Fig. 15G, H). A similar increased colocalization was observed in the BM of tPA-deficient mice with B-ALL (Fig. 4C; Supplementary Fig. 15I, J).

Treatment with IGF1 modestly increased the numbers of BCR-ABL1[+] BA/F3, but not MLL-AF9[+] BA/F3 cells (Fig. 4D). This effect was blocked in BCR-ABL1[+] BA/F3 cells using the mTOR inhibitor KU-0063794 (Supplementary Fig. 15K). These data suggest that growth factors such as IGF1 colocalize with ECM proteins in the BM of ANXA2 KO or tPA KO compared to WT mice and may be trapped there. Decreased mTORC2 levels within the IGF1 signaling pathway may lead to decreased numbers and/or survival of BCR-ABL1[+] cells.

## Differential sensitivity of leukemias to signaling events downstream of IGF1R

We then hypothesized that there is differential oncogene- and/or lineage-dependent sensitivity of leukemias to mTORC2 signaling downstream of the IGF1R, which may explain the observed phenotypes of CML, B-ALL and AML in ANXA2 KO mice. Indeed, IGF1 stimulated an increase in protein levels of rapamycin-insensitive companion of mammalian target of rapamycin (Rictor), one of the main components of mTORC2[36], and an increase of pAKT[S473] in BCR-ABL1[+] BA/F3 (Fig. 5A) and primary murine B-ALL cells (Fig. 5B), but not in empty vector[+], MLL-AF9[+] (BA/F3) or primary murine MLL-AF9[+] AML counterparts. In contrast, the regulatory-associated protein of mTOR (Raptor), a component of mTOR complex 1 (C1), and pS6K1, downstream of mTORC1, were not affected by IGF1 treatment (Supplementary Fig. 16A, B). Similarly, highest levels of mTORC2, particularly Rictor and pAKT[S473], were seen in K562 (BCR-ABL1[+] CML cell line) and NALM6 cells (B-ALL cell line), compared with the AML cell lines THP1 and Kasumi (Supplementary Fig. 16C). However, IGF1 treatment resulted

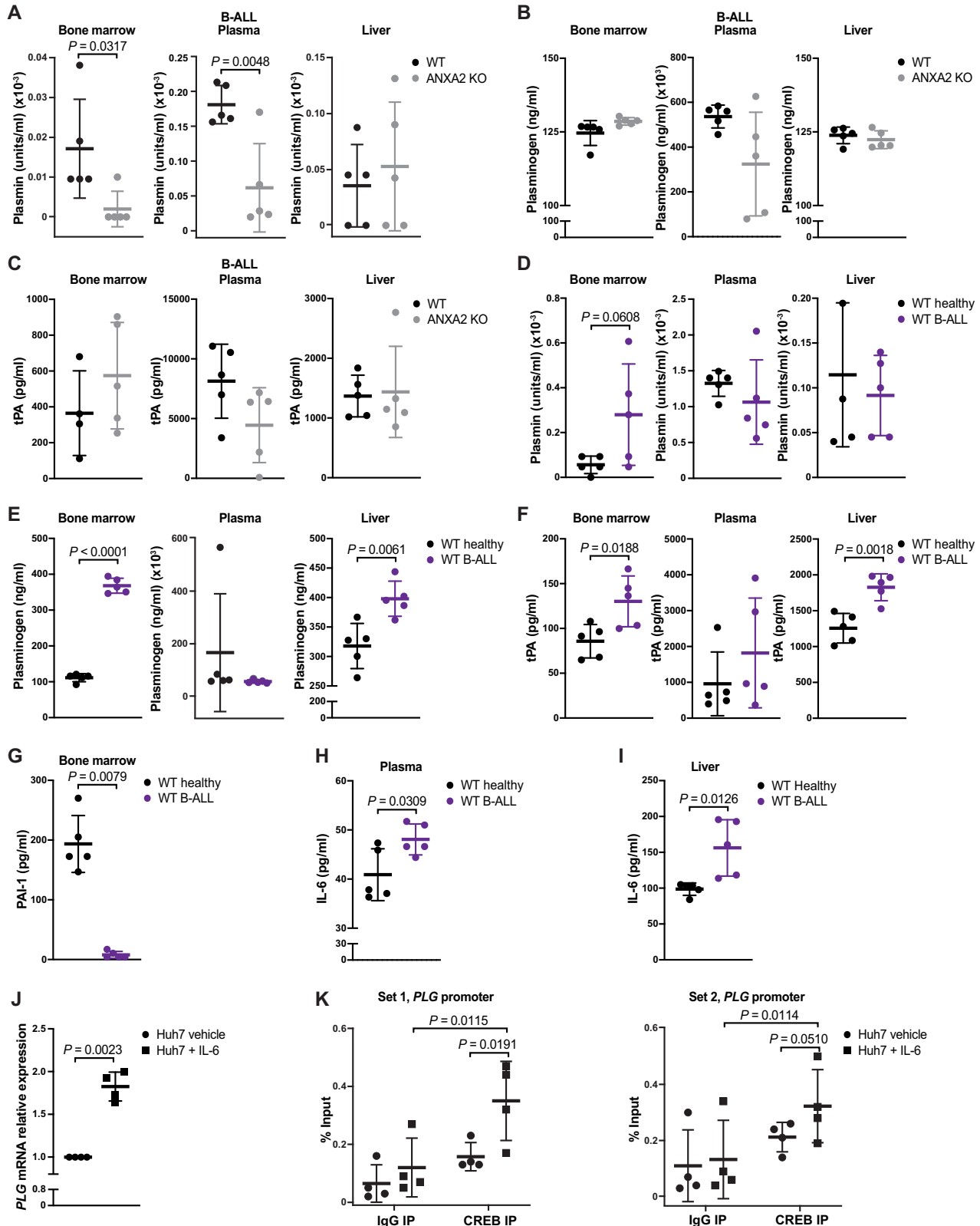

in increased levels of both Rictor and pAKT$^{S473}$ only in NALM6 cells (Fig. 5C). Consistently, highest levels of IGF1R were found on leukemia cells from mice with BCR-ABL1$^+$ B-ALL (Fig. 5D) and BCR-ABL1$^+$ BA/F3 cells (Supplementary Fig. 16D) compared to MLL-AF9$^+$ cells. Testing differences between the BCR-ABL1$^+$ cell lines K562 and SUPB15, whereby the former is of myeloid and the latter of lymphoid lineage, we found higher expression of IGF1R on

SUPB15 cells (Fig. 5E). After stimulation with IGF1 Rictor and—as a trend—pAKT$^{S473}$ were higher in SUPB15 compared to K562 cells (Fig. 5F). Taken together, these data suggest that IGF1 differentially affects signaling in BCR-ABL1$^+$ versus MLL-AF9$^+$ cells, whereby the effect is greatest on BCR-ABL1$^+$ lymphoid compared to BCR-ABL1$^+$ myeloid cells, possibly due to higher expression of IGF1R on B-ALL cells.

**Fig. 3 | B-ALL promotes hepatic generation of plasminogen. A** Active plasmin in supernatants of BM, plasma or liver from WT (black) or ANXA2 KO (gray) recipient mice with B-ALL ($P = 0.0317$, two-tailed Mann–Whitney test, $n = 5$ mice per group, mean ± SD). **B** Plasminogen in supernatants of BM, plasma or liver from WT (black) or ANXA2 KO (gray) recipient mice with B-ALL ($n = 5$ mice per group, mean ± SD). **C** Tissue plasminogen activator (tPA) in supernatants of BM, plasma or liver from WT (black) or ANXA2 KO (gray) recipient mice with B-ALL (WT $n = 4$, ANXA2 KO $n = 5$, mean ± SD). **D** Active plasmin in supernatants of BM, plasma or liver from healthy WT (black) or WT recipient mice with B-ALL (purple) ($P = 0.0608$, two-tailed $t$ test, $n = 5$ mice per group, mean ± SD). **E** Plasminogen in supernatants of BM, plasma or liver from healthy WT (black) or WT recipient mice with B-ALL (purple) ($P < 0.0001$, $P = 0.0061$, two-tailed $t$ test $n = 5$ mice per group, mean ± SD). **F** tPA in supernatants of BM, plasma or liver from healthy WT (black) or WT recipient mice with B-ALL (purple) ($P = 0.0188$, $P = 0.0018$, two-tailed $t$ test $n = 5$ mice per group,

mean ± SD). **G** Plasminogen-activator inhibitor 1 (PAI-1) in the BM supernatant from healthy WT mice (black) or WT recipient mice with B-ALL (purple) ($P = 0.0079$, two-tailed Mann–Whitney test, $n = 5$ mice per group, mean ± SD). Mice in (**A**)–(**G**) were unirradiated. **H**, **I** Interleukin 6 (IL-6) in plasma (**H**) and liver (**I**) from healthy WT (black) or WT recipient mice with B-ALL (purple) ($P = 0.0309$, $P = 0.0126$, two-tailed $t$ test, $n = 5$ mice per group, mean ± SD). **J** Relative expression of plasminogen (*PLG*) in Huh7 cells treated with vehicle (circles) or IL-6 (squares) ($P = 0.0023$, two-tailed $t$ test, $n = 4$ biological replicates, mean ± SD). **K** CREB binding to the promoter of *PLG* in Huh7 cells treated with vehicle (circles) or IL-6 (squares) in a chromatin immunoprecipitation (ChIP) assay ($P = 0.0191$, $P = 0.0510$, two-way ANOVA, Sidak test, $n = 4$ biological replicates, mean ± SD). We cannot exclude that the employed antibody to plasminogen may also be detecting plasmin. Source data are provided as a Source Data file.

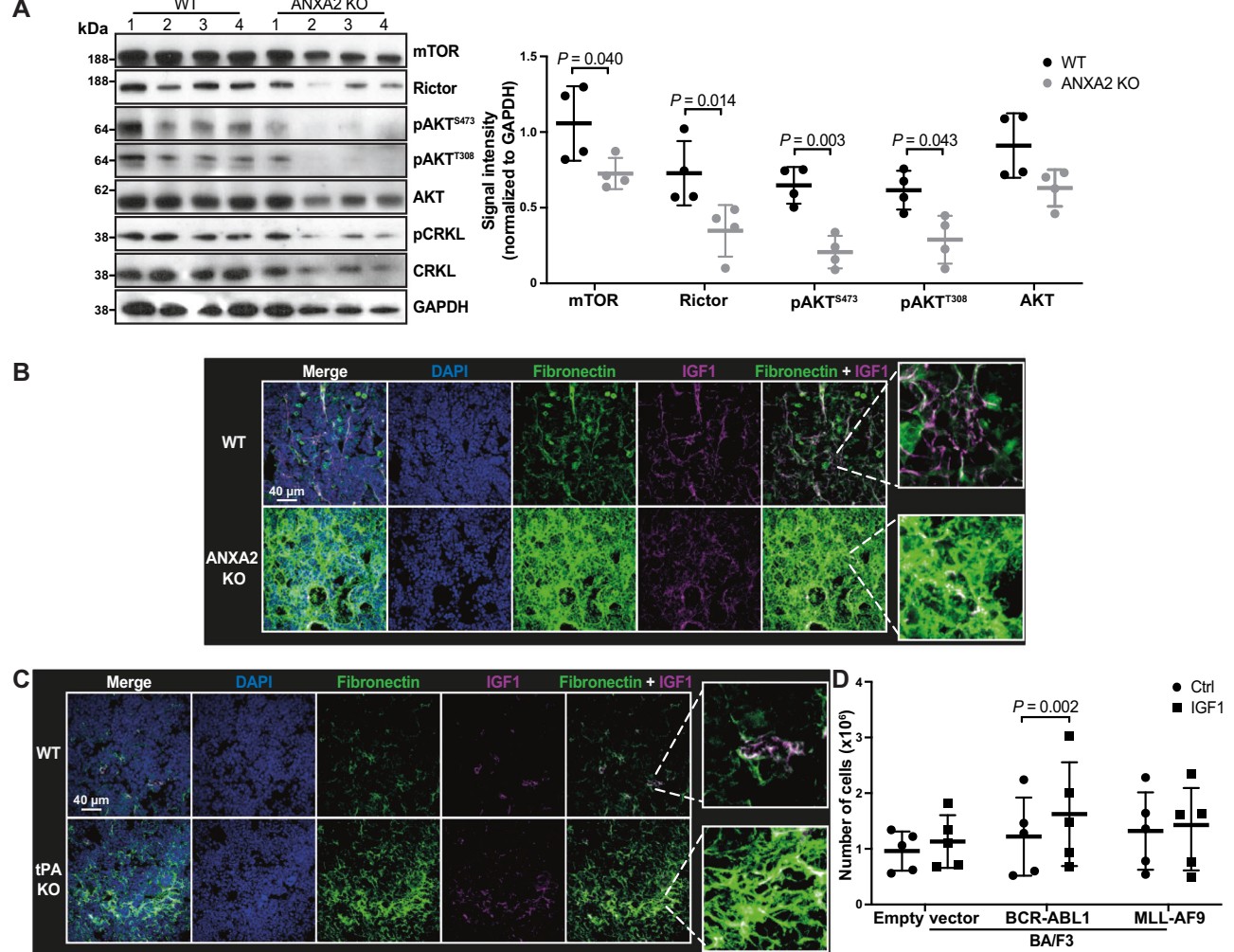

**Fig. 4 | Restricted access to growth factors in the BMM leads to reduced mTORC2 activation and proliferation of BCR-ABL1⁺ cells. A** Immunoblot for the indicated proteins in lysates of sorted GFP (BCR-ABL1)⁺ BP1⁺ BM cells from WT (black) or ANXA2 KO (gray) recipient mice with BCR-ABL1⁺ B-ALL. Each column represents a single mouse ($P = 0.04$, $P = 0.014$, $P = 0.003$, $P = 0.043$, two-way ANOVA, Sidak test, $n = 4$ mice per group, mean ± SD). **B** Immunofluorescence of bones, stained for fibronectin (green), DAPI (blue) and insulin-like growth factor

(IGF)1 (purple), from WT or ANXA2 KO recipient mice with BCR-ABL1⁺ B-ALL ($n = 3$ mice per group). **C** Immunofluorescence of bones, stained for fibronectin (green), DAPI (blue) and IGF1 (purple), from WT or tPA KO recipient mice with BCR-ABL1⁺ B-ALL ($n = 3$ mice per group). **D** Number of empty vector⁺, BCR-ABL1⁺ or MLL-AF9⁺ BA/F3 cells after 48 h of culture in medium containing vehicle (circles) or murine recombinant IGF1 (squares) (12 ng/ml) ($P = 0.002$, two-way ANOVA, Sidak test, $n = 5$ biological replicates, mean ± SD). Source data are provided as a Source Data file.

## Plasmin regulates IGF1 availability by contributing to breakdown of the ECM in an ANXA2-dependent manner

Hypothesizing that MSC-mediated plasmin activation may regulate IGF1 availability, we found that primary BCR-ABL1⁺ B-ALL cell numbers significantly increased when IGF1 was released from matrigel containing WT MSC after plasmin activation by plasminogen (Fig. 6A). This increase of BCR-ABL1⁺ B-ALL cell numbers was due to a decrease of late apoptosis, when the BCR-ABL1⁺ B-ALL cells were cultured on WT, but not ANXA2 KO MSC embedded in matrigel containing IGF1 and PLG (Supplementary Fig. 17). However, in absence of ANXA2 in

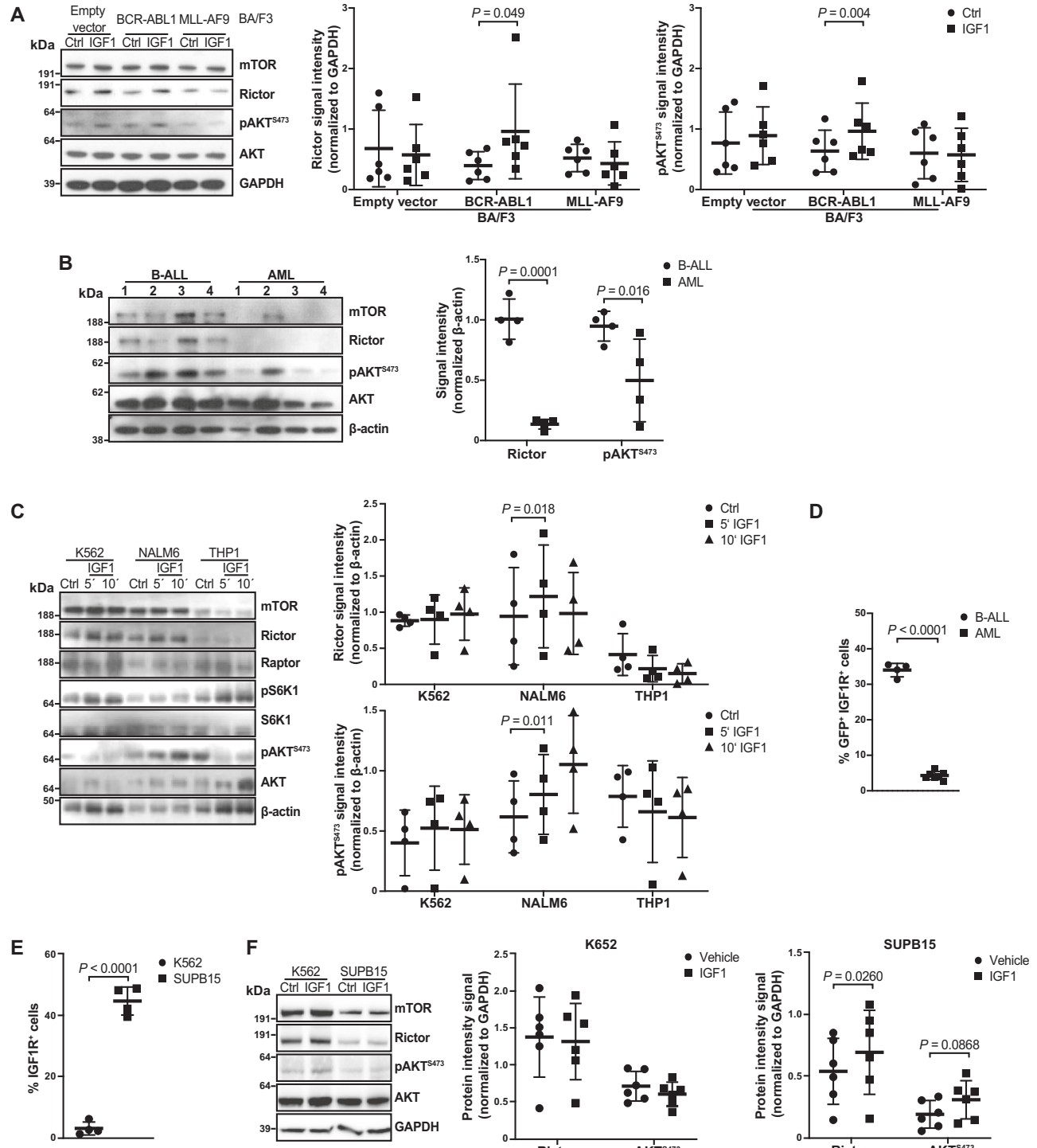

**Fig. 5 | Differential sensitivity of leukemias to signaling events downstream of IGF1R. A** Immunoblot for the indicated proteins in lysates of empty vector⁺, BCR-ABL1⁺ and MLL-AF9⁺ BA/F3 cells treated with vehicle (circles) or IGF1 (squares) and quantification (right) of the band intensity for Rictor and pAKT^S473 normalized to GAPDH ($P = 0.049$, $P = 0.004$, two-way ANOVA, Sidak test, $n = 6$ biological replicates, mean ± SD). **B** Immunoblot for the indicated proteins in lysates of sorted GFP (BCR-ABL1 or MLL-AF9)⁺ BM cells from individual WT recipient mice with BCR-ABL1⁺ B-ALL (circles) or MLL-AF9⁺ AML (squares) and quantification of the band intensity for Rictor and pAKT^S473 normalized to β-actin ($P = 0.0001$, $P = 0.016$, two-way ANOVA, Sidak test, $n = 4$ biological replicates, mean ± SD). **C** Representative immunoblot for the indicated proteins in lysates of K562, NALM6 or THP1 cells treated with vehicle (circles) or for 5 (squares) or 10 (triangles) minutes human recombinant IGF1 (12 ng/ml) and its quantification (right) of the band intensities for Rictor and pAKT^S473 normalized to β-actin ($P = 0.018$, $P = 0.011$, two-way ANOVA,

Dunnett test, $n = 4$ biological replicates, mean ± SD). The samples are derived from the same experiment, but one gel was used for mTOR, Rictor, pAktS473, Akt and β-actin. Another gel was used for Raptor, pS6K1 and S6K1. **D** Percentage of IGF1-receptor (IGF1R)⁺ cells of all GFP (BCR-ABL1)⁺ B-ALL (circles) versus all GFP (MLL-AF9)⁺ AML cells (squares) in the murine system ($P < 0.0001$, two-tailed $t$ test, B-ALL $n = 4$, AML $n = 7$ biological replicates, mean ± SD). **E** Percentage of IGF1-receptor (IGF1R)⁺ of all K562 (circles) versus SUPB15 (squares) cells ($P < 0.0001$, two-tailed $t$ test, $n = 4$ biological replicates, mean ± SD). **F** Representative immunoblot for the indicated proteins in lysates of K562 and SUPB15 cells treated with vehicle (circles) or human recombinant IGF1 (squares) and quantification (right) of the band intensities for Rictor and pAKT^S473 normalized to GAPDH ($P = 0.026$, $P = 0.0868$, two-way ANOVA, Sidak test, $n = 6$ biological replicates, mean ± SD). Source data are provided as a Source Data file.

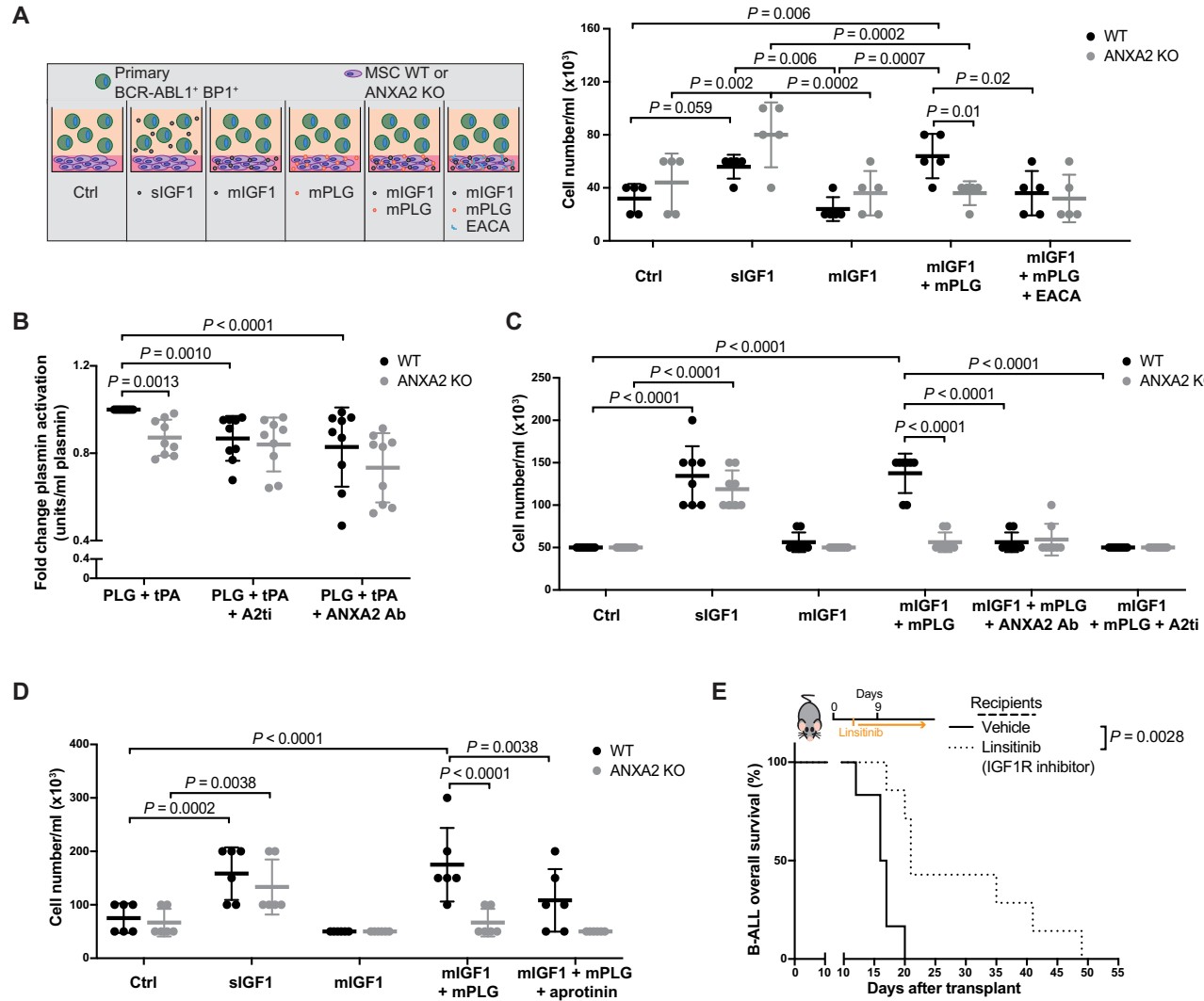

**Fig. 6 | Plasmin regulates IGF1 availability by contributing to breakdown of the ECM in an ANXA2-dependent manner. A** Number of sorted primary GFP (BCR-ABL1)⁺ BP1⁺ cells from the BM of WT mice with BCR-ABL1-induced B-ALL. Leukemia cells were plated on either WT (black) or ANXA2 KO (gray) MSC embedded in matrigel in different conditions (schematic on left). IGF1 was added in solution (sIGF1) or embedded in matrigel (mIGF1). mPLG = plasminogen (1 ng/ml) and EACA = ε-aminocaproic acid were embedded in matrigel (two-way ANOVA, Tukey test, *n* = 5 biological replicates, mean ± SD). **B** Active plasmin in the medium of primary WT (black) or ANXA2 KO (gray) mesenchymal stromal cells (MSC) in presence or absence of the ANXA2/S100A10 heterotetramer inhibitor (A2ti) (50 μm) or an antibody against ANXA2 (ANXA2 Ab) (60 μg/ml) (*P* = 0.0013, *P* = 0.001, *P* < 0.0001, two-way ANOVA, Tukey test, *n* = 9 biological replicates, mean ± SD), normalized to WT. **C** Number of sorted primary GFP (BCR-ABL1)⁺ BP1⁺ cells from the BM of WT mice with BCR-ABL1-induced B-ALL. Leukemia cells were plated on WT

(black) or ANXA2 KO (gray) MSC embedded in matrigel in different conditions. IGF1 was added in solution (sIGF1) or embedded in matrigel (mIGF1). mPLG = plasminogen, ANXA2 Ab (50 μm) and A2ti (60 μg/ml) were embedded in matrigel (two-way ANOVA, Tukey test, *n* = 8 biological replicates, mean ± SD). **D** Number of sorted primary GFP (BCR-ABL1)⁺ BP1⁺ cells from the BM of WT mice with BCR-ABL1-induced B-ALL. Leukemia cells were plated on WT (black) or ANXA2 KO (gray) MSC embedded in matrigel in different conditions. IGF1 was added in solution (sIGF1) or embedded in matrigel (mIGF1). mPLG = plasminogen and aprotinin were embedded in matrigel (two-way ANOVA, Tukey test, *n* = 6 biological replicates, mean ± SD). **E** Kaplan–Meier-style survival curve of WT recipient mice with BCR-ABL1⁺ B-ALL treated with vehicle (solid line) or the insulin-like growth factor receptor 1 inhibitor linsitinib (dotted line) (*P* = 0.0028, Log-rank test, vehicle *n* = 6, linsitinib *n* = 7). Source data are provided as a Source Data file.

MSC or in presence of the plasmin activation inhibitor ε-aminocaproic acid (EACA) no increase of B-ALL cell numbers was observed (Fig. 6A). In order to test the relevance of plasmin for breakdown of the ECM and subsequent release of IGF1 from the ECM, we, firstly, showed that plasminogen activation was reduced in ANXA2 KO MSC, in WT MSC treated with an inhibitor of the ANXA2/S100 A10 heterotetramer (A2ti)[37] or treated with an antibody that blocks the tPA-binding domain of ANXA2 (ANXA2 Ab)[38] (Fig. 6B). Secondly, addition of A2ti or ANXA2 Ab to matrigel containing WT MSC, IGF1 and PLG prevented an increase of primary BCR-ABL1⁺ B-ALL cells via IGF1 release from the matrix (Fig. 6C). And thirdly, plasmin inhibition by aprotinin significantly decreased BCR-ABL1⁺ B-ALL cell numbers cultured on top of

an ECM containing IGF1 and PLG (Fig. 6D). Lastly, treatment of WT mice with B-ALL with an inhibitor of IGF1R significantly extended survival (Fig. 6E). In sum, these data suggest that ANXA2 is involved in degradation of the ECM via its mediation of plasminogen activation. Breakdown of the ECM, in turn, releases growth factors such as IGF1, which promote the proliferation of B-ALL cells.

## Targeting fibrinolysis-associated pathways may extend survival of leukemic mice

We next analyzed therapeutic implications by treating mice with EACA, an approved anti-hemorrhagic drug. Indeed, treatment of non-leukemic mice with EACA led to a significant increase in

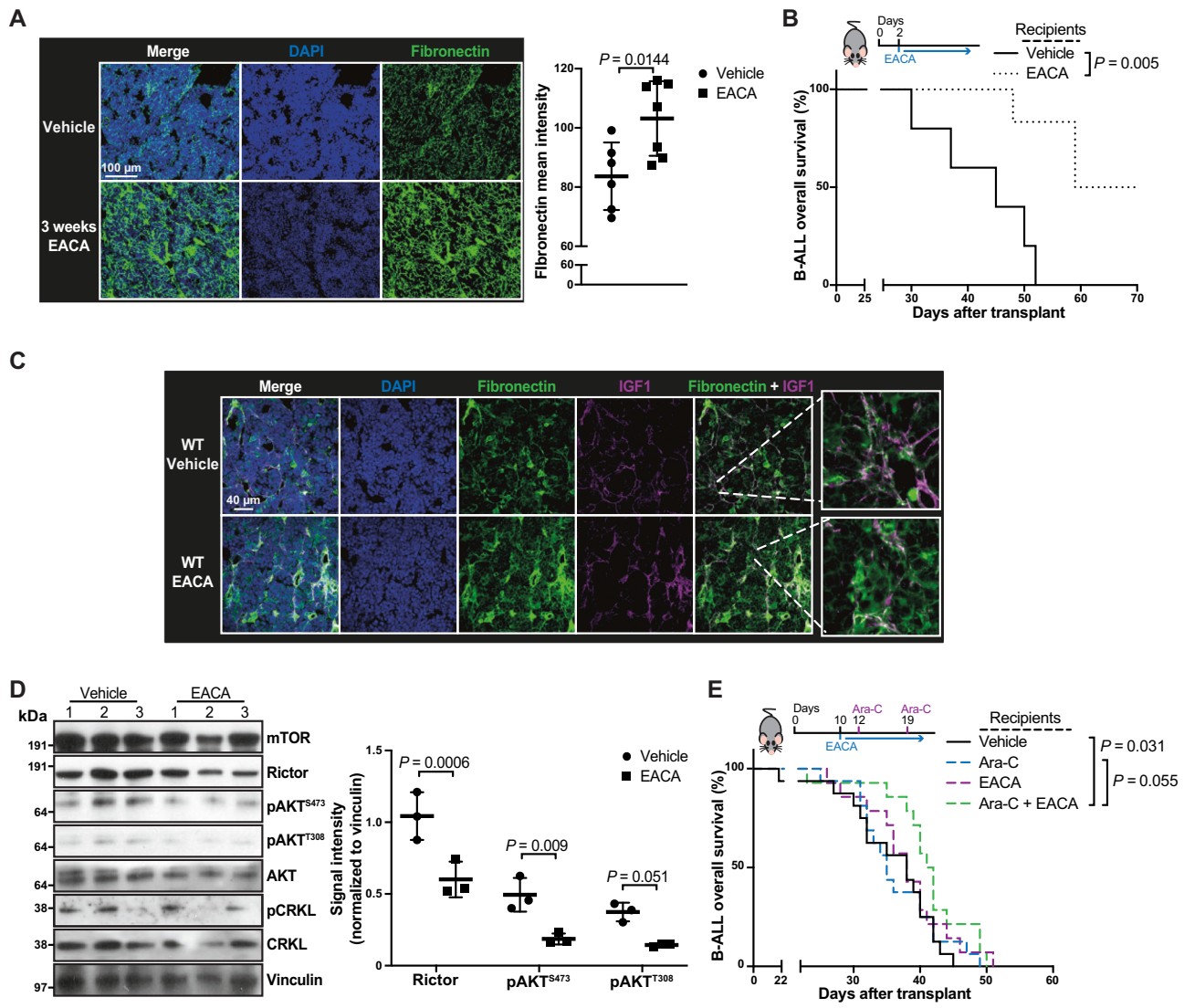

**Fig. 7 | Targeting fibrinolysis-associated pathways may extend survival of leukemic mice. A** Immunofluorescence of bones of normal WT mice treated with vehicle (circles) or ε-aminocaproic acid (EACA) (squares) (1.2 mg/kg daily for 3 weeks), stained for fibronectin (green) and DAPI (blue) ($P = 0.0144$, two-tailed $t$ test, $n = 3$ mice (with 2–3 images per mouse), mean ± SD). **B** Kaplan–Meier-style survival curve of WT recipient mice with BCR-ABL1[+] B-ALL treated with vehicle (solid line) or ε-aminocaproic acid (EACA; dotted line) (1.2 mg/kg daily) starting from day 2 after transplant ($P = 0.005$, Log-rank test, vehicle $n = 5$, EACA $n = 6$). **C** Immunofluorescence of bones from WT recipient mice with BCR-ABL1[+] B-ALL treated with vehicle or EACA starting from day 2 after transplant, as in (**B**), stained for fibronectin (green), IGF1 (purple) and DAPI (blue) ($n = 3$ mice pre group).

**D** Immunoblot analysis of sorted GFP (BCR-ABL1)[+] BP1[+] BM cells from WT recipient mice with BCR-ABL1[+] B-ALL treated with vehicle (circles) or EACA (squares) starting from day 2 after transplantation as in (**B**). The quantification on the right shows the band intensity of Rictor, pAKT[S473] and pAKT[T308] normalized over the loading control vinculin (two-way ANOVA, Sidak test, $n = 3$ mice per group, mean ± SD). **E** Kaplan–Meier-style survival curve of WT recipient mice with BCR-ABL1[+] B-ALL treated with vehicle (black), cytarabine (ara-C; blue), EACA (purple) or a combination of ara-C and EACA (green) as from day 10 after transplantation ($P = 0.031$, $P = 0.055$, Log-rank test, vehicle $n = 14$, ara-C $n = 16$, EACA $n = 15$ and ara-C and EACA $n = 14$). The analyses in (**C**, **D**) were performed on day 20 after transplantation. Source data are provided as a Source Data file.

fibronectin levels in the BMM (Fig. 7A). Treatment of WT mice with BCR-ABL1[+] B-ALL with EACA after presumably completed homing[1,39], led to significant reduction of the GFP (BCR-ABL1)[+] BP1[+] tumor load (Supplementary Fig. 18A), leukemic infiltration into various organs (Supplementary Fig. 18B–H) and survival extension (Fig. 7B). ANXA2-deficient mice were resistant to these effects (Supplementary Fig. 18I). EACA treatment of mice with BCR-ABL1[+] B-ALL resulted in significantly higher fibronectin levels as seen in ANXA2-deficient mice (Fig. 7C; Supplementary Fig. 18J). B-ALL cells from the BM of EACA-treated mice also showed a significant reduction in the levels of Rictor and pAKT[S473] (Fig. 7D). Combination treatment with EACA and low dose cytarabine (ara-C), considered standard therapy in B-ALL and initiated when disease was detectable (Supplementary

Fig. 18K), resulted in a significant survival extension compared to mice receiving vehicle (Fig. 7E, Supplementary Fig. 18L). Lastly, intravenous treatment of mice with B-ALL with EACA significantly reduced levels of plasmin/-ogen, plasminogen and tPA in the BM compared to vehicle treatment (Supplementary Fig. 19A–C). Local administration of EACA to the BMM by intrafemoral injection led to increased fibronectin levels in the BMM of WT, but not ANXA2 KO mice (Supplementary Fig. 19D). These data support the concept that targeting plasminogen activation via administration of EACA may represent an adjunct therapy in B-ALL. While the beneficial effect of EACA in the BMM was also achieved by systemic administration, intrafemoral administration of EACA directly increased fibronectin levels in the BMM.

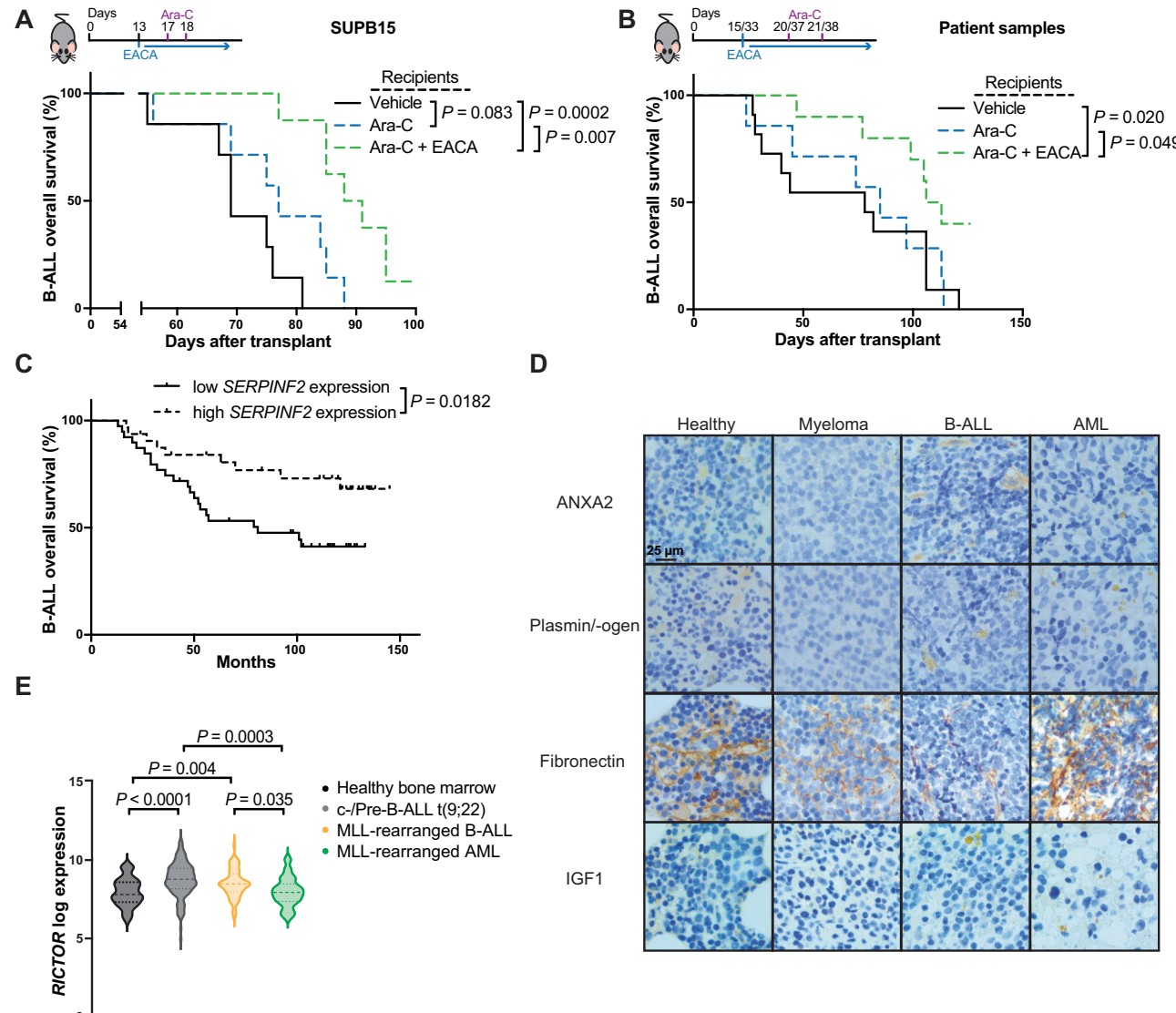

**Fig. 8 | Fibrinolysis-associated pathways may influence leukemia progression in human patients. A** Kaplan–Meier-style survival curve of NOD SCID interleukin-2 receptor γ knockout (NSG) mice transplanted with SUP-B15 cells (BCR-ABL1+), treated with vehicle (black), ara-C (blue) or with a combination of ara-C and EACA (green) ($P = 0.0002$, $P = 0.007$, Log-rank test, vehicle $n = 7$, ara-C $n = 7$, ara-C and EACA $n = 8$). **B** Kaplan–Meier-style survival of NSG mice transplanted with leukemia cells from 6 patients with B-ALL. Each sample was transplanted into at least 3 mice, whereby one mouse was treated with vehicle (black), one with ara-C (blue) and one with a combination of ara-C and EACA (green). Treatment started on day 15 or day 33 after transplantation depending on the successful detection of hCD45+ cells in the peripheral blood of transplanted mice after transplantation ($P = 0.02$, $P = 0.049$, Log-rank test, vehicle $n = 8$, ara-C $n = 7$, ara-C + EACA $n = 10$). If enough B-ALL cells

were available, more mice were transplanted. **C** Kaplan–Meier-style survival of patients with B-ALL with high ($n = 32$) or low ($n = 39$) expression of *SERPINF2* (α2-antiplasmin). The curves were generated using data from the publicly available dataset TARGET (phase II), which was accessed via the cBioPortal ($P = 0.0182$, Log-rank test). **D** Immunohistochemistry of bones from healthy individuals, patients with multiple myeloma, B-ALL and AML stained for the indicated proteins ($n = 5$ per condition). **E** Log2 expression of *RICTOR* in BM cells of healthy individuals ($n = 73$; black), patients with c-/pre-B-ALL positive for the BCR-ABL1 oncoprotein (t(9;22)) ($n = 122$; gray), MLL-rearranged pro-B-ALL ($n = 73$; yellow) or MLL-rearranged AML ($n = 38$; green), taken from the BloodSpot portal (MILE study) (two-way ANOVA). Source data are provided as a Source Data file.

## Fibrinolysis-associated pathways may influence leukemia progression in human patients

To test whether our findings in mice may be transferrable to human leukemia cell lines, we transplanted the BCR-ABL1+ B-ALL cell line SUP-B15 into NOD SCID interleukin-2 receptor γ knockout mice (NSG), which were treated with vehicle, ara-C or the combination of ara-C plus EACA. This revealed a reduction of the human CD45+ CD19+ tumor burden in peripheral blood (Supplementary Fig. 20A), a significant increase of fibronectin levels in the BMM of ara-C plus EACA-treated mice (Supplementary Fig. 20B) and survival extension in mice treated with the combination of ara-C plus EACA compared to treatment with ara-C alone (Fig. 8A). Transplantation of primary human B-ALL cells

into at least three NSG mice per donor, whereby each mouse was randomly assigned to the treatment groups vehicle, ara-C alone or ara-C plus EACA, led to significant reduction of the percentage of human CD45+ CD19+ cells in PB (Supplementary Fig. 20C, D), significant extension of survival (Fig. 8B) and significantly increased levels of fibronectin in the BMM in most xenotransplanted mice (Supplementary Fig. 20E) in the double-treated cohort compared to mice treated with ara-C alone. In contrast, treatment of NSG mice transplanted with THP1 (AML) cells with ara-C plus EACA did not impact survival (Supplementary Fig. 20F).

Hypothesizing that higher levels of the main natural inhibitor of plasmin, α2-antiplasmin (encoded by the *SERPINF2* gene), in B-ALL

cells may influence survival in human patients, we analyzed publicly available datasets[40–44]. In line with the mouse experiments, high *SERPINF2* expression in leukemia cells was associated with a significant survival extension in patients with B-ALL (Fig. 8C; Supplementary Fig. 20G). In addition, *SERPINF2* expression in leukemia cells was significantly lower in patients with B-ALL compared to AML (Supplementary Fig. 20H), but significantly higher in BCR-ABL1[+] B-ALL cells versus healthy BM (Supplementary Fig. 20I). Although patient numbers are too small to draw definitive conclusions, immunohistochemistry of bone sections of patients with B-ALL, AML or multiple myeloma (the latter acting as control) may tentatively support our murine data with regards to expression of ANXA2 (Fig. 8D; Supplementary Fig. 21A), plasmin/-ogen (Fig. 8D; Supplementary Fig. 21B), fibronectin and IGF1 (Fig. 8D; Supplementary Fig. 21C, D) in some B-ALL samples compared to multiple myeloma. However, fibronectin staining in B-ALL, but not AML, seemed to significantly correlate with IGF1 in the same patients (Fig. 8D; Supplementary Fig. 21E, F). Further analysis of publicly available datasets[40,43,44] revealed higher expression of *RICTOR* in patients with (BCR-ABL[+]) B-ALL compared to healthy controls and AML, including AML patients with MLL rearrangements (Fig. 8E; Supplementary Fig. 21G). Taken together, these data suggest the concept that BMM-associated ANXA2/plasmin/IGF1-signaling may also be important for human B-ALL, but larger studies are needed to address this.

## Discussion

Here we show that the fibrinolytic pathway including ANXA2 plays a differential role for progression of BCR-ABL1[+] B-ALL versus MLL-AF9[+] AML. We implicate plasmin-mediated degradation of the ECM in the BMM and IGF1 release from this ECM in the regulation of B-ALL expansion. In turn, via TNFα[19,30] B-ALL cells condition MSC and possibly other cells of the BMM to secrete IL-6, which, in turn, activates hepatocytes to produce plasminogen.

Whether B-ALL progression was impaired in an ANXA2-deficient BMM, characterized by a denser ECM than in WT mice, due to entrapment of growth factors alone or, additionally, due to mechanical hindrance of B-ALL progression could not be completely clarified. However, we did observe decreased homing of B-ALL LIC in an ANXA2 KO BMM and more pronounced survival extension in unirradiated ANXA2 KO recipients, suggesting that mechanical or irradiation-associated factors on the ECM may be contributory, as described[45]. In support of this and our work, irradiation has been shown to reduce the stiffness of an in vitro collagen matrix[46], which may lead to reduced cancer spread and prolonged survival. However, irradiation effects may be dependent on the dominant ECM protein in the respective tissue[47]. Albeit, in our model quantitative irradiation-induced ECM changes were ruled out. Additionally, endothelial cells were increased in healthy, unirradiated ANXA2 KO mice, suggesting that decreased angiogenesis in unirrradiated ANXA2 KO mice, as it has been described[48], is less likely to be the reason for prolonged B-ALL survival in this model.

While our studies were largely focused on MSC, we show that at least monocytes/macrophages[14,15], endothelial cells, major producers of tPA, for instance in vascular niches of the BMM or hepatic sinusoids, and fibroblasts participate in plasminogen activation, but contributions of other cell types cannot be excluded. Furthermore, consistent with the increase of plasminogen in B-ALL, uPAR, an important component of the plasminogen activation system and contributor to ECM lysis[49], was also increased. Future studies are needed to address the role of this and the PAR pathways to B-ALL.

In various solid cancers[50,51], expression of ANXA2 has been correlated with invasion, metastasis, resistance to treatments and decreased patient survival via regulation of plasmin-dependent degradation of the ECM[52], consistent with our study. Our work extends knowledge of ECM remodeling and release of cytokines and

GFs in bone metastasis of solid tumors[53] to leukemia, where ECM proteins are known to influence disease progression[19,54,55].

In line with a previously published report on the leukemia-specific influence of the BMM on leukemic course[1], we show here that plasmin-mediated release of IGF1 specifically mediates pro-survival signaling in B-ALL cells via stimulation of the mTORC2/AKT pathway, as previously shown[56]. In CML, this effect was less pronounced, possibly reflecting differences between BCR-ABL1[+] myeloid and lymphoid leukemias or differences between CML in chronic versus more advanced stages[57,58] with regards to IGF1/mTORC2 signaling. Other cytokines entrapped in the ECM[18] may be contributing to the observed survival extension in CML in ANXA2 KO mice. In contrast, in MLL-AF9[+] AML IGF1 activates mTORC1, and mTORC1 inhibition impairs AML progression[59]. Little is known on IGF1-mediated activation of mTORC2 signaling in AML[60], but we show that IGF1R expression is diminished on MLL-AF9[+] AML cells, suggesting that AML cells may be less sensitive to variation in IGF1 levels and mTORC2 signaling. But whether these findings are restricted to MLL-AF9[+] AML or are applicable to other AML subtypes needs to be studied in the future.

In PML-RARα-associated APML expression of the ANXA2/S100A10 complex causes hyperfibrinolysis due to the accumulation of plasmin on the surface of leukemia cells[7]. The contribution of ANXA2/S100A10 from the BMM, however, had not been investigated previously, although the in vitro migration of T-ALL cell lines or T-cell lymphoma growth[25] are dependent on PLG.

Lastly, our results suggest that manipulation of the levels of ECM proteins and of GF availability by EACA in addition to standard chemotherapy may efficiently reduce tumor burden or extend the survival of mice with B-ALL. Given the approved use of EACA in the hemorrhagic setting, administration of EACA to B-ALL patients who are frequently thrombocytopenic and at risk of bleeding, would likely be safe.

In conclusion, our data demonstrate that ANXA2 and the associated fibrinolytic pathway have highly differential roles for progression of BCR-ABL1[+] B-ALL and CML versus MLL-AF9[+] AML. Adjunct treatments targeting plasmin activation, to be tested in clinical trials, raise hopes for consistently successful treatment of B-ALL in future.

## Methods

All research complied with relevant ethical regulations and approved by the local government (Regierungspräsidium Darmstadt and the Landesuntersuchungsamt Rheinland-Pfalz, Germany) for murine studies and the ethics committees of the respective universities for human studies.

All authors complied with the guidelines on inclusion and ethics in global research.

### Mice

5–6-week- or 8–10-week-old C57/BL6 mice were purchased from Charles River Laboratories (Sulzfeld, Germany) and were used as wildtype (WT) donors or WT recipients, respectively, in all transplants. 8–10-week-old C57/BL6 mice were also used for the isolation of primary mesenchymal stromal cells (MSC), fibroblasts and macrophages. ANXA2 knockout (KO) (C57/BL6 background) mice[16] were a kind gift from Prof. Katherine Hajjar and bred in the animal facility of the Georg-Speyer-Haus, Institute for Tumor Biology and Experimental Therapy. They were used at an age of 8–10 weeks. 5–6-week-old tissue plasminogen activator (tPA) KO recipients (C57/BL6 and SJL background mix) were purchased from Molecular Innovation (Novi, MI, USA). The respective control mice (B6SJLF1/J) were purchased from The Jackson Laboratory (Bar Harbor, ME, USA) and used at an age of 5–6 weeks as donors and at an age of 8–10 weeks as recipients. NOD SCID interleukin (IL)-2 receptor γ knockout (NSG) mice were bred in the inhouse animal facility. Animals of different sexes and ages were randomly assigned to experimental groups. All mouse strains had been backcrossed for at least 7 generations. All animal studies were approved by the local

government (Regierungspräsidium Darmstadt and the Landesuntersuchungsamt Rheinland-Pfalz, Germany). The maximally allowed tumor burden of 80% (oncogene⁺ Gr1⁺, oncogene⁺ CD11b⁺ or oncogene⁺ BP1⁺ cells) was not exceeded. Mice were sacrificed according to the criteria stated in the animal studies approved by the local government. The mice were housed in individually ventilated cages at an ambient temperature between 20–26 degrees Celsius. A 14-h light/10 h dark cycle was used.

## Human samples

Studies on cryopreserved unsorted cells from bone marrow or peripheral blood from patients with B-ALL were approved by the local ethics committee bylaws in the hematology departments of the Hospital Lyon Sud, Pierre Bénite and Centre Leon Bérard, Lyon, France. The use of bone sections from patients with B-ALL, AML and multiple myeloma was approved by the Ethics Committee of the University Clinic of the Goethe University Frankfurt (Approval number 274/18 and SHN-5-2020). The age or gender of the patients, as well as the genetic subtype of the leukemia, were unknown, consistent with the ethics committee's approval. Each patient had signed an informed consent. No compensation was provided to the patients for the inclusion of their samples in this study.

## Cell lines

The human cell line 293T (ACC635) and the mouse cell line NIH/3T3 (ACC59) were purchased from the German Collection of Microorganisms and Cell Cultures (DSMZ) and cultured in DMEM, 10% fetal bovine serum (FBS), 1% penicillin/streptomycin and 1% L-glutamine. The medium for 293T cells was further supplemented with 1% non-essential amino acids. The murine cell line H5V (a kind gift from Prof. Stefanie Dimmeler) was cultured in DMEM, 10% fetal bovine serum (FBS), 1% penicillin/streptomycin and 1% L-glutamine. The mouse cell line MS5 (ACC441) was purchased from the DSMZ and was cultured in α-MEM with 10% FBS, 1% penicillin/streptomycin and 1% L-glutamine. The murine pro-B cell line BA/F3 (ACC300) was purchased from the DSMZ and grown in RPMI, 10% fetal bovine serum (FBS), 1% penicillin/streptomycin and 1% L-glutamine supplemented with 5% (v/v) WEHI medium as a source of interleukin 3 (IL-3). Cells were transduced on two consecutive days with cryopreserved MSCV-IRES-GFP (empty vector), MSCV-IRES GFP BCR-ABL1- (as in vitro model for murine B-ALL) or MSCV-IRES GFP MLL-AF9- (as in vitro model for murine AML) expressing retroviri. The human leukemic cell lines K562 (BCR-ABL1⁺; myeloid, model of CML) (ACC10), NALM6 (lymphoid, model of BCR-ABL1-negative B-ALL) (ACC128), THP1 (MLL-AF9⁺; myeloid, model of AML) (ACC16), SUPB15 (lymphoid, model of BCR-ABL1-positive B-ALL) (ACC389) and Kasumi (AML1-ETO⁺; myeloid, model of AML) (ACC220) were purchased from the DSMZ and cultured in RPMI, 10% FBS, 1% penicillin/streptomycin and 1% L-glutamine. These cell lines were used to test the activation of mTORC2 in response to stimulation with IGF1, dependent on oncogene or lymphoid versus myeloid lineage. The human hepatocellular carcinoma cell line, Huh7, was a kind gift from Prof. Ivan Dikic and was cultured in DMEM, 10% fetal bovine serum (FBS), 1% penicillin/streptomycin and 1% L-glutamine. The Huh7 cell line was used as in vitro model to test the production of PLG by hepatic cells in response to different conditions. Primary mouse BCR-ABL1⁺ B-ALL cells were cultured in RPMI, 10% fetal bovine serum (FBS), 1% penicillin/streptomycin and 1% L-glutamine supplemented with IL-7 (10 ng/ml). Primary murine MSC were grown in MEM, 20% FBS, 1% penicillin/streptomycin and 1% L-glutamine. 100 µg/ml primocin (InvivoGen, Toulouse, France) was added to the culture medium until passage two. All cell lines were maintained in a 37 °C, 5% $CO_2$ incubator. All cell lines were routinely tested for mycoplasma contamination and tested negative.

## Microarray

Acute myeloid leukemia (AML) and chronic myeloid leukemia (CML) were induced in Col2.3 GFP reporter mice. On day 45 for AML and day 19 for CML, mice were sacrificed, and long bones were collected and crushed. Bones were further digested in 1.8 mg/ml collagenase type I (Worthington, Biochemical Corporation, Lakewood, NJ) solution. Total RNA from 40,000–50,000 sorted GFP⁺ Ter119-PE⁻ CD45-PE⁻ osteoblastic cells was extracted using RNeasy Plus Micro Kit (Qiagen, Germantown, MD) according to the manufacturer´s protocol. RNA was amplified using Ovation Pico WTA reagent (NuGEN, Redwood City, CA). Amplified cRNA was labeled with the GeneChip wild-type terminal labeling kit (Affymetrix, Santa Clara, CA), hybridized to Mouse Gene ST 2.2 SST microarrays (Affymetrix, Santa Clara, CA), and scanned by GeneChip Scanner 3000 7G system (Affymetrix, Santa Clara, CA) according to standard protocols. The primary microarray data was normalized and analyzed using R with limma and lumi packages[61,62]. False discovery rate (FDR, cut off analysis <0.1) was used for multiple test correction.

## Retrovirus production

To generate MSCV IRES GFP- (empty vector), MSCV IRES GFP BCR-ABL1- or MSCV IRES GFP MLL-AF9-expressing retroviri, 293 T cells were co-transfected with the respective plasmids (10 µg/3 × 10⁶ cells) and ecopak plasmid (5 µg/3 × 10⁶ cells). 48 h after transfection the conditioned medium containing the retroviri was harvested, and viral titers were tested via the transduction of NIH/3T3 cells.

## Bone marrow transduction/transplantation

To induce CML, AML or the MSCV empty vector control condition, donor BM cells were harvested from 5-fluorouracil (5-FU)-pretreated mice (200 mg/kg), which had received 5-FU 4 days prior to harvest. Subsequently, BM cells were prestimulated overnight with stem cell factor (Peprotech, Hamburg, Germany) (50 ng/ml), interleukin (IL)-3 (Peprotech, Hamburg, Germany) (6 ng/ml) and IL-6 (Peprotech, Hamburg, Germany) (10 ng/ml)[1]. Cells were subsequently transduced on two consecutive days with cryopreserved MSCV IRES GFP BCR-ABL1 (CML)- or MLL-AF9 (AML)-expressing retrovirus or MSCV IRES GFP empty vector and intravenously (i.v.) transplanted (2.5 × 10⁵ cells/mouse for CML and 5 × 10⁵ cells/mouse for AML) into sublethally irradiated (2 × 450 cGy) mice. For induction of B-cell acute lymphoblastic leukemia (B-ALL) BM cells were harvested from non-5-FU-treated mice, transduced once with MSCV IRES GFP BCR-ABL1-expressing retrovirus and transplanted i.v. (1 × 10⁶ cells/mouse) into sublethally irradiated (2 × 450 cGy) or into non-irradiated (2 × 10⁶ cells/mouse) mice.

For the secondary transplants cells were harvested from the bones or spleens of mice with CML (sacrificed on day 14) or B-ALL (sacrificed on day 20). PB analysis prior to sacrifice confirmed full establishment of the disease. In the CML and B-ALL secondary transplantation models 3 × 10⁶ BM or spleen cells were intravenously transplanted into each sublethally irradiated (2 × 450 cGy) WT secondary recipient.

For the xenotransplantation experiments, 1 × 10⁶ SUPB15 cells (human BCR-ABL1⁺ B-ALL cell line) were injected into non-irradiated NSG mice, or (1.5–2) × 10⁶ cells from the BM or PB of six different patients with B-ALL were intravenously transplanted into at least 3 sublethally irradiated (200 cGy) NSG mice. Individual mice from each group were randomly assigned to treatment with vehicle, cytarabine (ara-C) (Sigma-Aldrich, Darmstadt, Germany) or the combination of ara-C and ε-aminocaproic acid (EACA) (Sigma-Aldrich, Darmstadt, Germany) (for treatment details see below).

## In vivo drug treatment

In the B-ALL rescue experiment ANXA2-deficient mice were treated once a week with intraperitoneal (i.p.) injection of vehicle (PBS) or plasmin (Sigma-Aldrich, Darmstadt, Germany) (1 mg/kg) starting 48 hrs after

transplantation. For the B-ALL treatment experiments vehicle (PBS) or EACA (Sigma-Aldrich, Darmstadt, Germany) (1.2 mg/kg) were injected subcutaneously (s.c.) on a daily basis, starting either on day 2 or 10 after transplantation (as indicated in the legends). Ara-C (0.5 mg/kg) was administered i.p. on days 12 and 19 after transplantation. Vehicle (oil) or linsitinib (inhibitor of insulin-like growth factor receptor 1) (MedChem-Express, Sollentuna, Sweden) were administered by oral gavage as a daily regimen (75 mg/kg starting from day 9-11 and 25 mg/kg starting from day 12). In intrafemoral (i.f.) experiments mice with B-ALL were treated with EACA (Sigma-Aldrich, Darmstadt, Germany) (1.2 mg/kg) by intrafemoral (i.f.) administration on day 15 after transplantation or tPA (Sigma-Aldrich, Darmstadt, Germany) (10 mg/kg) by i.f. administration on day 19. 24 h after treatment mice were sacrificed. In the xeno-transplantation experiments NSG mice were subcutaneously (s.c.) treated with EACA (1.2 mg/kg) or vehicle (PBS) daily starting on day 13 after transplantation for experiments involving SUPB15 cells and on day 33 after transplantation when human primary B-ALL cells were used. Ara-C was administered i.p. (75 mg/kg) on days 17 and 18 after transplantation (SUPB15 experiment) and on days 20/21 or 37/38 after transplantation (human primary B-ALL cells), respectively.

### Homing experiment
To assess homing in the B-ALL model, BM cells were harvested from non-5-FU-treated mice and transduced once with MSCV IRES GFP BCR-ABL1-expressing retrovirus. After overnight culture in RPMI, 10% fetal bovine serum (FBS), 1% penicillin/streptomycin and 1% L-glutamine supplemented with IL-7 (Peprotech, Hamburg, Germany) (10 ng/ml), $5 \times 10^6$ cells/mouse were intravenously transplanted into sublethally irradiated ($2 \times 450$ cGy) mice. 18 h after transplantation BM and spleen cells from recipient mice were analyzed for the percentage of GFP (BCR-ABL1)$^+$ BP1$^+$ cells. A list of all antibodies can be found in Supplementary Table 1.

### Analysis of diseased mice and tumor burden
Disease progression in leukemic mice was assessed by analysis of peripheral blood using a Scil Vet animal blood counter (Scil Animal Care Company, Viernheim, Germany). In parallel, flow cytometry (BD Fortessa, Heidelberg, Germany) was used to determine the tumor burden represented by the percentage of GFP (BCR-ABL1)$^+$ leukocytes positive for allophycocyanin (APC)-conjugated CD11b antibody (CML) or phycoerythrin (PE)-conjugated BP-1 antibody (B-ALL) and the percentage of GFP (MLL-AF9)$^+$ leukocytes positive for allophycocyanin-Cy7 (APC-Cy7)-conjugated Gr1 antibody (AML). PE-conjugated human CD45 and APC-conjugated CD19 antibodies were used to assess the tumor burden in xenotransplanted NSG mice. A list of all antibodies can be found in Supplementary Table 1. Flow cytometric analyses were performed using the FlowJo software (version 10). Gating strategies can be found in Supplementary Fig. 22.

### Progenitor analysis in normal and leukemic mice
Bones, spleen and peripheral blood were harvested from normal WT or ANXA2-deficient mice or from mice transplanted with BM transduced with BCR-ABL1- (to induce CML) or empty vector-expressing retrovirus. Bones and spleens were crushed in PBS containing 2% FBS. Subsequently, cells were stained for the lineage markers CD11b, Ter119, CD5, B220 and F4/80 to identify Lin$^-$ cells. Multipotent progenitors (MPP) were identified as Lin$^-$ Thy1.1$^-$ Sca1$^+$ c-Kit$^+$ Flk2$^+$ and common lymphoid progenitors (CLP) were identified as Lin$^-$ IL7Rα$^+$ Thy1.1$^-$ Sca-1$^{lo}$ c-Kit$^{lo}$. A list of all antibodies can be found in Supplementary Table 1.

### Colony-formation assay
$2 \times 10^4$ total BM or $2 \times 10^5$ spleen cells from individual mice with CML (sacrificed on day 15 after transplantation) were plated in triplicate in cytokine-free methylcellulose medium (M3234, Stemcell Technologies, Cologne, Germany) supplemented with 100 pg/ml IL-3

(Peprotech, Hamburg, Germany). For the B-ALL model, $2 \times 10^5$ total BM or $2 \times 10^5$ spleen cells from individual mice with B-ALL (sacrificed on day 20 after transplantation) were plated in triplicate in methyl-cellulose medium for pre-B lymphoid progenitor cells (M3630, STEMCELL Technologies, Cologne, Germany). In all conditions colonies were scored after 7 days.

### Immunohistochemistry
Bones were collected from diseased mice and fixed in 10% formalin overnight at room temperature. After decalcification for 1–2 weeks in 0.5 M EDTA bones were mounted in paraffin and sectioned. Immuno-histochemistry of bone sections from mice or humans was performed using antibodies for fibronectin, ANXA2, plasmin/-ogen and insulin-like growth factor (IGF1) according to standard procedures. A list of all antibodies can be found in Supplementary Table 1. Staining was detected by immunoperoxidase using a yellow-brown horseradish-peroxidase chromogen.

### Isolation of GFP$^+$ stroma cells from the BMM
To isolate stroma cells from the bone marrow microenvironment (BMM), femora, tibiae, pelvis, spine and humeri were harvested from nestin-GFP (GFP$^+$ mesenchymal stromal cells), Col2.3-GFP (GFP$^+$ osteoblastic cells) or Tie2-GFP (GFP$^+$ endothelial cells) mice. Bones were crushed in PBS containing 2% FBS and stained with lineage markers (see above). Lin$^+$ cells were depleted with streptavidin beads (Miltenyi Biotec, Bergisch Gladbach, Germany). The Lin$^-$ fraction was further stained with antibodies to CD31, except when sorting for endothelial cells, as well as CD45 and Ter119. A list of all antibodies can be found in Supplementary Table 1. Cells positive for GFP, but negative for the markers above were sorted directly into RLT plus buffer (Qia-gen, Düsseldorf, Germany) or lysed with RIPA buffer (50 mM Tris HCl pH 7.4, 150 mM NaCl, 1% Triton X-100, 1% Na DOC, 0.1% SDS, 1 mM EDTA). Gating strategies can be found in Supplementary Fig. 22.

### Isolation of primary stromal cells
Long bones from WT or ANXA2-deficient mice were harvested, cut into small pieces and digested for 1 h at 37 °C in collagenase. Subsequently, CD45$^-$ Ter119$^-$ PDGFRα$^+$ Sca1$^+$ MSC were sorted (BD FACSAria Fusion Cell Sorter, Franklin Lakes, NJ, USA) and cultured until passage six[63]. A list of all antibodies can be found in Supplementary Table 1. Gating strategies can be found in Supplementary Fig. 22.

For the isolation of primary macrophages (MΦ), long bones from WT or ANXA2-deficient mice were harvested and crushed in phosphate-buffered saline (PBS) using mortar and pestle. After 1 week non-adherent cells were removed and adherent cells were cultured in αMEM medium supplemented with 20% fetal bovine serum, 1% peni-cillin/streptomycin and 1% L-glutamine. The immunophenotype was confirmed using F4/80$^+$ CD169$^+$ to identify macrophages.

For the isolation of fibroblasts, long bones from WT or ANXA2-deficient mice were harvested. After flushing, bones were cut into pieces, distributed on the plate and cultured in enough DMEM medium supplemented with 10% FBS, 1% penicillin/streptomycin and 1% L-glutamine to cover the pieces. After 1–2 weeks, when enough adherent cells were spreading out of the pieces of bones, the pieces were removed. The immunophenotype was confirmed by positivity for αSMA and vimentin to identify fibroblasts.

Primary macrophages and primary fibroblasts were used to repeat some of the experiments performed with primary MSC to test, whether the role of ANXA2 in the BMM is specific to MSC. For the isolation of osteoblasts long bones from WT mice were harvested, cut into small pieces and digested for 1 h at 37 °C in collagenase. Subsequently, CD4$^-$ CD8$^-$ Ter119$^-$ CD11b$^-$ Gr1$^-$ CD31$^-$ Sca1$^-$ CD51$^+$ osteoblastic cells were sorted (BD FACSAria III Cell Sorter, Franklin Lakes, NJ, USA), and RNA isolation was performed. A list of all antibodies can be found in Sup-plementary Table 1.

For the isolation of endothelial cells long bones from WT mice were harvested, cut into small pieces and digested for 1 h at 37 °C in collagenase, as above. Subsequently, depletion of CD45+ cells was performed using CD45 MicroBeads (Miltenyi Biotec, Bergisch Gladbach, Germany). Endothelial cells were then enriched by positive selection of CD31+ cells using CD31 MicroBeads (Miltenyi Biotec, Bergisch Gladbach, Germany). Cells were resuspended in TRIzol reagent (ThermoFisher, Darmstadt, Germany) to isolate RNA. These cells were used to test the levels of ANXA2 in different stromal cell populations.

### Differentiation and colony-forming unit fibroblast (CFU-F) assays of MSC

$1 \times 10^4$ WT or ANXA2-deficient MSC were plated in 48-well plates. Cells were differentiated into adipocytes using the adipocyte differentiation and maintenance medium (LONZA, Cologne, Germany). After 10-14 days cells were stained with oil red O to identify adipocytes. To assess differentiation into osteoblasts, cells were cultured in αMEM containing 20% FBS, 1% penicillin/streptomycin and 1% L-glutamine supplemented with 50 μg/ml ascorbic acid, 10 mM β-glycerophosphate and 10 mM dexamethasone. After 10–14 days cells were stained with alizarin red S solution to assess osteoblast differentiation.

For the CFU-F assay $2 \times 10^3$ WT or ANXA2-deficient MSC were plated in 10 cm plates. Cells were cultured for 2 weeks in αMEM, 20% FBS, 1% penicillin/streptomycin and 1% L-glutamine. Twice a week half the volume of the medium was carefully removed and substituted with fresh medium. After 2 weeks cells were fixed with 4% paraformaldehyde (PFA) for 15 min at room temperature (RT) and stained with 1% crystal violet solution (in 20% methanol) at room temperature for 30 min. The colonies per plate were counted.

### RNAseq

Primary MSC was sorted from WT or ANXA2-deficient mice and collected in TRIzol reagent (ThermoFisher, Darmstadt, Germany). Chloroform (1/5th of TRIzol volume) was added to allow RNA extraction, and the RNA was further purified using an mRNA purification kit (Qiagen, Düsseldorf, Germany) following the manufacturer's protocol.

cDNA was generated and amplified using 8.7 ng of total RNA and the SMART-Seq® v4 Ultra® Low Input RNA Kit for Sequencing (Takara) following the manufacturer's protocol. Then cDNA was sheared with the Covaris ultrasonicator followed by preparation of the sequencing library using NEBNext End Repair module, dA-Tailing module and Quick Ligation module (New England BioLabs).

The final libraries were quality controlled by Agilent 4200 TapeStation System (Agilent Technologies) and Qubit ds DNA HS Assay kit (Life Technologies-Invitrogen). Based on Qubit quantification and sizing analysis, sequencing libraries were normalized, pooled and clustered on the cBot (Illumina) with a final concentration of 250 pM (spiked with 1% PhiX control v3). 100 bp paired-end sequencing was performed on the Illumina HiSeq 4000 instrument using standard Illumina protocols.

Adapter sequences were trimmed from paired end RNA-seq reads using TrimGalore (https://github.com/FelixKrueger/TrimGalore). Quality-filtered reads were mapped using STAR[64] against the mouse genome (GRCm39). Read counts were quantified with HTSeq[65], and differential expression analysis was carried out with these counts using Bioconductor package DESeq2[66]. Volcano plots were plotted using an R package EnhancedVolcano (https://github.com/kevinblighe/EnhancedVolcano).

### Co-immunoprecipitation

MSC was lysed in RIPA buffer (50 mM Tris HCl pH 7.4, 150 mM NaCl, 1% Triton X-100, 1% Na DOC, 0.1% SDS, 1 mM EDTA). Equal amounts of protein per sample were incubated with 2 μg of antibody against S100A10 (Invitrogen, Darmstadt, Germany) for 3 h at 4 °C followed by overnight incubation with magnetic beads (Dynabeads Protein G,

ThermoFisher Scientific, Darmstadt, Germany) at 4 °C. After several washes with ice cold RIPA buffer the protein-bead complex was resuspended in Laemmli elution buffer and heated to 95 °C for 5 min. The eluted proteins were run on an SDS page gel, as described below.

### Immunoblotting

Cultured mouse or human cells were stimulated with mouse or human recombinant insulin-like growth factor (IGF) 1 (12 ng/ml), respectively, for 5 min. Cultured cells or primary BM or spleen cells were lysed using RIPA buffer (50 mM Tris HCl pH 7.4, 150 mM NaCl, 1% Triton X-100, 1% Na DOC, 0.1% SDS, 1 mM EDTA), freshly supplemented with protease and phosphatase inhibitor cocktails (Sigma-Aldrich, Darmstadt, Germany). Lysates were kept on ice for 1 h and centrifuged at $15,000g$ at 4 °C for 20 min. Protein concentrations were measured using the Bradford protein assay (Bio-Rad, Hercules, CA, USA). Equal amounts of protein were mixed with 4x Laemmli buffer, supplemented with β-mercaptoethanol, denatured and run on either 4-12% Bis-Tris polyacrylamide gels (ThermoFisher Scientific, Darmstadt, Germany) or custom-made gels containing 10% acrylamide. Proteins were blotted on PVDF membranes (ThermoFisher Scientific, Darmstadt, Germany). Subsequently, the membranes were probed overnight at 4 °C in Tris-buffered saline, 0.1% Tween 20 (TBST) with antibodies to pAKT$^{S473}$, pAKT$^{T308}$, pCRKL, and pS6K1 (1:1000), in 2.5% BSA with antibodies to mTOR, Rictor and Raptor (1:1000) or in 5% milk with antibodies to ANXA2, S100A10, CRKL, AKT and S6K1 (1:1000), or GAPDH, β-actin and vinculin (1:2000). A list of all antibodies can be found in Supplementary Table 1. After incubation with secondary horseradish-peroxidase (HRP)-conjugated antibodies, membranes were developed using X-ray films (Fujifilm, Düsseldorf, Germany). Band intensities were quantified using Image J software.

### Plasmin and MMP-9 activation assay

To test plasmin activation, cells, which had been plated at $5 \times 10^3$ cells/well (96-well plate), were pre-incubated with plasminogen (PLG) (Sigma-Aldrich, Darmstadt, Germany) (170 nM) for 1 h. Then tissue plasminogen activator (tPA) (Sigma-Aldrich, Darmstadt, Germany) (12 nM) was added to the medium covering WT or ANXA2 KO MSC. After 5 min of incubation at 37 °C plasmin substrate (D-Val-Leu-Lys-4-nitroanilid-dihydrochlorid) (Sigma-Aldrich, Darmstadt, Germany) was added to the culture medium, and after 10 min the production of the cleaved colorogenic substrate, indicative of active plasmin, was measured at 405 nm. To test plasmin-dependent activation of matrix metalloproteinase (MMP)-9, a 24-h-pretreatment of MSC with tumor necrosis factor (TNF)α (Peprotech, Hamburg, Germany) (15 ng/ml) in 1% FBS medium was followed by addition of tissue plasminogen activator (tPA) (Sigma-Aldrich, Darmstadt, Germany) and plasminogen (PLG) (Sigma-Aldrich, Darmstadt, Germany). To assess MMP-9 activity, 100 μl of conditioned medium was mixed with 100 μl of 2x assay buffer (400 nM NaCl, 100 mM Tris-HCL pH7.6, 10 mM CaCl2, 40 μM ZnSO4, 0.1% Brij 35) and incubated for 1 h at 37 °C with MMP-9 fluorogenic substrate (10 μM) (Calbiochem, Darmstadt, Germany). Cleavage of the substrate was measured at 365 nm[19]. Only the MMP-9-activation assay was performed in 1% FBS. All the other assays were performed in the absence of serum.

### CRISPR

The sgRNAs were designed to target the sequence of *Anxa2* using the Benchling design tool (https://benchling.com/crispr) (Supplementary Table 3). The primers were cloned into lentiCRISPR.v2 (Addgene plasmid #52961; http://n2t.net/addgene:52961; RRID:Addgene_52961)[67] using BsmBI sites. A non-targeting (NTC) sgRNA was used as control and cloned into both vectors. Integration was confirmed by sequencing using the LKO.1 primer shown in Supplementary Table 3.

The ANXA2 KO cells were generated by transducing MS5, H5V and NIH/3T3 cells with lentivirus expressing the sgRNA. After transduction,

cells were cultured in 10 μg/ml puromycin to select for transduced cells. The KO efficiency was tested by immunoblot analysis.

## Immunofluorescence

To test fibronectin or laminin degradation, $1 \times 10^4$ WT or ANXA2-deficient MSC were cultured in a 48-well plate for one week to allow deposition of fibronectin or laminin. After addition of tPA (Sigma-Aldrich, Darmstadt, Germany) (0.7 ng/ml) and PLG (Sigma-Aldrich, Darmstadt, Germany) (1 ng/ml) to the medium and culture for 6 h at 37 °C, cells were fixed in 4% PFA (Santa Cruz, Heidelberg, Germany) for 10 min at room temperature (RT). After 30 min of blocking with 2% BSA at RT samples were incubated overnight with a primary antibody to fibronectin or laminin (Abcam, Cambridge, UK) (1:200 in 2% BSA). The following day samples were incubated with fluorophore-labeled secondary antibody (Alexa Fluor 647, Invitrogen, Darmstadt, Germany) for 1 h at RT, and nuclei were counterstained with 5 μg/ml 4′,6-diamidin-2-phenylindol (DAPI). Images were acquired using the confocal quantitative image cytometer CQ1 (Yokogawa, Germany), and fibronectin staining was quantified using Image J software.

## Immunofluorescence of bone sections

Bones were harvested from normal or diseased mice and fixed with 4% PFA overnight at 4 °C. Following decalcification in 0.5 M EDTA for 1–2 week bones were rehydrated in 2% polyvinylpyrrolidone (PVP) (Sigma-Aldrich, Darmstadt, Germany) overnight at 4 °C, frozen in Tissue-Tek optimum cutting temperature O.C.T. (Sakura, Alphen aan den Rijin, The Netherlands) and stored at −80 °C. Sections were 10 μm thick. After thawing slides were blocked for 30 min with 2% BSA and incubated overnight at 4 °C with primary antibodies to fibronectin and IGF1 (1:200 in 2% BSA) (Abcam, Cambridge, UK) or laminin (1:200 in 2% BSA) (Abcam, Cambridge, UK). The following day slides were incubated with the secondary antibodies Alexa Fluor 647 or 488 (Invitrogen, Darmstadt, Germany) at a 1:400 dilution for 1 h at room temperature. Nuclei were counterstained with DAPI (5 μg/ml). Images were acquired using the confocal quantitative image cytometer CQ1 (Yokogawa, Germany) and staining was quantified using Image J software.

## Invasion assay

$1 \times 10^4$ WT or ANXA2-deficient MSC, respectively, were mixed in a solution of 0.5 mg/ml matrigel (Corning, Wiesbaden, Germany) ± PLG (Sigma-Aldrich, Darmstadt, Germany) (1 ng/ml) and plated on top of a transwell (3 μm pore size, Corning, Wiesbaden, Germany). Upon matrigel polymerization $2.5 \times 10^5$ BCR-ABL1$^+$ BA/F3 cells were plated on top of the matrigel in RPMI containing 2% FBS. RPMI containing 20% FBS was added to the lower chamber. After 18 h cells, which had invaded into the lower chamber, were counted.

## Plasmid for ANXA2 overexpression

The *Anxa2* cDNA template was amplified from NIH/3T3 cells by PCR and cloned as GST-fusion into the bacterial expression vector pGEX-6P1. The insert was then recloned into a mammalian expression vector similar to the commercial vector pEGFP-C1 (Clontech) with the sole difference that the EGFP sequence was replaced by the HA-tag sequence. The resulting vector was named 'pHA-Annexin A2, mouse'.

## Transient ANXA2 overexpression

ANXA2-deficient MSC was plated in a 6-well plate and transduced with the ANXA2-expressing plasmid (1.5 μg) using lipofectamine 2000 (ThermoFisher, Darmstadt, Germany) according to the manufacturer's protocol. 48 h later the transduction efficiency was tested by qRT-PCR and immunoblot, and cells were used in the invasion assay.

## Quantitative real-time PCR

Samples were collected in TRIzol reagent (ThermoFisher, Darmstadt, Germany). Chloroform (1/5th of TRIzol volume) was added to allow RNA extraction, and the RNA was further purified using an mRNA purification kit (Qiagen, Düsseldorf, Germany) following the manufacturer's protocol. Equal amounts of RNA, measured with a NanoDrop (ThermoFisher, Darmstadt, Germany), were reverse transcribed into cDNA using the ProtoScript First Strand cDNA Synthesis kit (New England Biolabs, Ipswich, MA). Quantitative RT-PCR was performed using SYBR green (ThermoFisher, Darmstadt, Germany). A list of primers can be found in Supplementary Table 2.

## Enzyme-linked immunosorbent assay (ELISA)

BM supernatant was collected by flushing a femur from each mouse with 300 μl of PBS. Liver and spleen supernatants were collected from each mouse by crushing the same amount (in g) of liver or spleen with 300 μl of PBS. The concentration of plasminogen, tissue plasminogen activator (tPA), plasminogen activator inhibitor- 1 (PAI-1), urokinase plasminogen activator surface receptor (uPAR), D-dimers, α2-antiplasmin, α2-macroglobulin and IL-6 in each sample was determined using the mouse plasminogen (Abcam, Cambridge, UK), mouse/rat tPA (Abcam, Cambridge, UK), the mouse PAI-1 (Invitrogen, Darmstadt, Germany), mouse uPAR (Abcam, Cambridge, UK), D-dimer (LSBio, MA, USA), α2-antiplasmin (LSBio, MA, USA), mouse α2-macroglobulin (Novus Biologicals, MN, USA) and mouse IL-6 (Invitrogen, Darmstadt, Germany) ELISA kits, respectively, following the manufacturer's protocol. The supernatant of Huh7 cells cultured in the conditioned medium of human MSC exposed to vehicle, TNFα (Peprotech, Hamburg, Germany) (15 ng/ml) or SUPB15 cells were collected after 6 h. The concentration of plasminogen in each sample was determined using the human plasminogen (Abcam, Cambridge, UK) ELISA kit following the manufacturer's protocol. We cannot exclude that the employed antibody to plasminogen may also be detecting plasmin.

## Chromatin immunoprecipitation (ChIP) assay

Huh7 cells treated with vehicle (water) or human recombinant IL-6 (Peprotech, Hamburg, Germany) (20 ng/ml) for 3 h, were treated with formaldehyde (0.75%). 1 M glycine was used to stop the reaction. Cells were collected in ChIP lysis buffer (50 mM HEPES KOH, 140 mM NaCl, 1% TritonX-100, 0.5 mM EDTA pH8, 0.1% sodiumdeoxycholate, 0.1% SDS) and incubated for 1 h at 4 °C followed by sonication to fragment the DNA. After removal of cellular debris (by centrifugation) the supernatant was snap-frozen in liquid nitrogen and stored at −80 °C. Antibodies to cAMP Responsive Element Binding Protein 1 (CREB1) or IgG control antibodies, together with protein A beads, previously blocked with RIPA buffer containing salmon sperm DNA (Sigma Aldrich, Darmstadt, Germany) (100 ng/μl) to prevent non-specific binding, were used for overnight immunoprecipitation at 4 °C. Several washes were then followed by DNA elution and incubation with proteinase K. The DNA was purified using the ChIP DNA Clean and Concentrator kit (ZymoResearch, Irvine, CA, USA) and eluted in 60 μl of elution buffer. qPCR analysis was performed using 2 μl of chromatin (DNA) with two different primer sets specific for the distal region of the plasminogen (*PLG*) promoter containing the CREB1 binding site. Primers were designed to recognize the CREB1 binding site on the promoter region of plasminogen using the Integrative genomics viewer (IGV) (https://software.broadinstitute.org/software/igv/) and are listed in Supplementary Table 4. The DNA recovery was normalized over the input (as percent of the input)[19].

## Proliferation assay

$2.5 \times 10^5$ empty vector$^+$, BCR-ABL1$^+$ or MLL-AF9$^+$ BA/F3 cells were cultured in RPMI containing 5% FBS with or without murine recombinant IGF1 (BioLegend, San Diego, CA, USA) (12 ng/ml), and after 48 h cell numbers were counted. Where indicated, KU-0063794 (Selleckchem, Munich, Germany) (5 μM) was added to the culture medium.

In the IGF1 release experiment $5 \times 10^3$ WT or ANXA2-deficient MSC were embedded in a matrigel solution (1.5 mg/ml) with or without PLG

(Sigma-Aldrich, Darmstadt, Germany) (1 ng/ml), EACA (Sigma-Aldrich, Darmstadt, Germany) (12.5 mM), ANXA2 antibody (Santa Cruz, Heidelberg, Germany) (60 μg/ml), ANXA2/S100A10 heterotetramer inhibitor (A2ti) (MedChemExpress, Sollentuna, Sweden) (50 μM) or aprotinin (Sigma-Aldrich, Darmstadt, Germany) (60 nM) and plated in a 48-well plate. IGF1 (12 ng/ml) was either added to the solution or the matrigel. After polymerization of the matrigel $(4-8) \times 10^4$ primary GFP (BCR-ABL1)$^+$ BP1$^+$ cells sorted from the BM of mice with BCR-ABL1-induced B-ALL were added to each condition. Cell were counted 48 h after plating.

To analyze apoptosis, primary BM cells were transduced with MSCV IRES GFP BCR-ABL1-expressing retrovirus, cultured overnight in RPMI, 10% fetal bovine serum (FBS), 1% penicillin/streptomycin and 1% L-glutamine supplemented with IL-7 (Peprotech, Hamburg, Germany) (10 ng/ml) and, finally, added to WT or ANXA2-deficient MSC embedded in a matrigel solution with PLG (Sigma-Aldrich, Darmstadt, Germany) (1 ng/ml) and IGF1 (12 ng/ml). Apoptosis was assessed by FACS after labelling the cells with Annexin 5 (BioLegend, San Diego, CA, USA) and 7-aminoactinomycin D (7-AAD) (Thermo-Fisher, Darmstadt, Germany). Annexin5$^+$ 7-AAD$^+$ cells were considered as late apoptotic cells. Gating strategies can be found in Supplementary Fig. 22.

### Whitlock-Witte limiting dilution assay

Primary BM cells from non-5-FU treated mice were transduced once with BCR-ABL1- expressing retrovirus and plated in triplicate in 24-well plates (on top of $1 \times 10^4$ WT or ANXA2-deficient MSC plated the night before) at the following dilutions: $1 \times 10^2$, $3 \times 10^2$, $1 \times 10^3$, $3 \times 10^3$, $1 \times 10^4$, $3 \times 10^4$, $1 \times 10^5$ and $3 \times 10^5$/well. Cells were cultured in RPMI containing 10% FBS, 200 μM L-glutamine, 50 μM β-mercaptoethanol and 1% penicillin/streptomycin (1 ml/well). Twice a week half of the medium was replaced with fresh medium. Cells were counted starting from day 5. Confluence was considered at a leukemia cell count of $10^6$/well[31].

### Statistics and reproducibility

All statistical analyses were performed using GraphPad software (Prism 8.0). Between three to six independent biological replicates were included in each experiment (except in Supplementary Fig. 10D). No statistical method was used to predetermine sample size. Data were excluded from the analysis only when they were defined as outlier by GraphPad software (Prism 8.0). The experiments were not randomized. The investigators were not blinded to allocation during experiments and outcome assessment.

Shapiro–Wilks normality test was applied to all experiments to test, if the data followed a normal distribution. If the data were normally distributed, a two-tailed t-test was used, when the means of two groups were being compared. When multiple hypotheses were tested, one-way or two-way ANOVA and Tukey or Sidak tests (the latter for multiple comparisons) were used as post-hoc tests. When the data did not follow a normal distribution, nonparametric tests were used. Two variables were compared using the unpaired, two-tailed Mann-Whitney test and more than two variables were compared by applying the Kruskal-Wallis test.

Survival curves were analyzed by Kaplan-Meier-style curves and Log-rank (Mantel-Cox) or Gehan-Breslow-Wilcoxon tests. In the Kaplan-Meier-style survival curve of patients with B-ALL, patients were stratified into high or low groups depending on whether their expression levels of *SERPINF2* in B-ALL cells were above or below the average of all patients in the dataset.

The data were presented as mean ± s.d (standard deviation). *P* values ≤ 0.05 were considered significant.

### Reporting summary

Further information on research design is available in the Nature Portfolio Reporting Summary linked to this article.

## Data availability

RNA sequencing and microarray data are available at Gene Expression Omnibus (GEO) under accession numbers GSE205762 and GSE205872, respectively. Data on the expression of genes in human samples and survival of patients with B-ALL were obtained from the publicly available dataset TCGA (TARGET phase II), accessed through the cBioPortal portal (https://www.cbioportal.org) and through the BloodSpot portal (https://www.fobinf.com/). Supplementary information is available for this paper. Correspondence and requests for materials should be addressed to the corresponding author. Source data are provided with this paper.

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

## Acknowledgements

This work was supported by grant 2015_A238 to D.S.K. from the Else Kröner-Fresenius-Stiftung and by the Discovery & Development Program grants 2020 and 2021 from the Frankfurt Cancer Institute (FCI) and Project number 449828949 from the Deutsche Forschungsgemeinschaft to V.R.M. The results on human data published here are in part based upon data generated by the Therapeutically Applicable Research to Generate Effective Treatments (https://ocg.cancer.gov/programs/target) initiative, phs000464. The data used for this analysis are available at https://portal.gdc.cancer.gov/projects. We thank the Genomics and Proteomics Core Facility of the DKFZ for performing the RNA-sequencing. The staining of bone sections was performed by the Department of Pathology of the University Medical Center Mainz.

## Author contributions

V.R.M. and D.S.K. designed the experiments. V.R.M., J.B. and C.K. performed the in vitro experiments. V.R.M., C.K., C.Z., R.K. and M.M. performed the in vivo experiments. R.P., N.T. and T.K. assisted with experiments, and P.L. assisted with CRISPR-CAS experiments. V.R.M. analyzed the data. M.P. and A.E. provided the ANXA2 overexpressing plasmid. S.H., K.B., V.M.-S. and S.L. provided the human samples. E.M. performed the bioinformatic analyses in the laboratory of B.J.P.H. W.R. provided input on experiments and critically reviewed the manuscript. V.R.M. wrote a first draft of the manuscript. D.S.K. supervised the research, analyzed data and wrote the manuscript.

## Funding

## Competing interests

The authors declare no competing interests.
