## [Transparent Peer Review file · Nature Communications]

Exploitation of the fibrinolytic system by B-cell acute lymphoblastic leukemia and its therapeutic targeting

Corresponding Author: Professor Daniela Krause

Version 0:

Reviewer comments:

Reviewer #1

(Remarks to the Author)

It has been known for some time that the fibrinolytic system, which generates the serine protease plasmin, promotes the release of hematopoietic stem cells from their bone marrow niche. In this interesting study, the authors provide substantial evidence for involvement of the fibrinolytic system, specifically its cell surface arm enabled by the annexin A2 (A2) complex, in remodeling of bone marrow extracellular matrix (BMM) and mobilization of BCR-ABL+ B-cell leukemia cells into the peripheral blood. The authors postulate that the A2 complex generates plasmin which liberates IGF1 from the BMM, allowing mTORC2-mediated signaling and proliferation of ALL cells. It is postulated further that ALL cells increase hepatic synthesis of plasminogen, the precursor of plasmin, creating a cyclical acceleration of the disease process. The authors suggest that adjunctive treatment with a fibrinolysis inhibitor, such as EACA, could reduce tumor burden and prolong survival.

The manuscript addresses a reasonable premise based on previously published literature. For the most part, the experiments appear to have been well-executed, though there are some issues to be resolved. The manuscript contains a large amount of data, much of which resides within the 20 supplemental figures. The work is original, and generally justifies the conclusions. There are, however, a few questions that should be resolved.

Major Concerns:

[1] Figure 1A and others. It is not clear why the authors rely upon fibronectin as a marker of BMM integrity? What is the evidence that fibronectin is a bona fide plasmin substrate in vivo? What about other BMM proteins, and what about fibrin, a primary plasmin substrate?

[2] Figure 2I and 2J. Systemic administration of plasmin. If a mouse weighs 30 gm, and the administered dose is 1 mg/kg, each mouse received around 30 ug of plasmin. In a blood volume of 1 ml, this concentration of plasmin (30 ug/ml) would likely be neutralized by its principal inhibitor, alpha 2-antiplasmin (~86 ug/ml) [Lijnen et al. Blood 93:2274, 1999], as well as other inhibitors, such as alpha2-macroglobulin.

[3] Figure 3, panels A, B, D, and E. Plasmin was measured as activity, and concentration estimated by standard curve. Plasminogen was measured by ELISA. To what extent does the plasminogen assay also measure plasmin?

[4] Suppl Fig 5E. Secondary survival rates are based on rather low numbers of mice (n=5-6). For the bone marrow result, loss of a single A2KO mouse out of 5 reduces survival by 20%. While this is technically a significant finding (p=0.03), it should be confirmed using larger numbers per group.

[5] Suppl Fig 5B and 5G. The authors provide evidence that the absence of A2 extends survival in CML, but not AML. What is the presumed mechanism for this based on the biology of the two disorders, and how do these results relate to the subsequent studies focusing on ALL?

[6] Suppl Fig 13A. n= 3-4 for animals receiving vehicle versus plasmin, and the p value seems to be non-significant (p=0.093). Also, what happens when one gives plasmin to WT animals with ALL? Is there accumulation of plasmin and

depletion of fibronectin?

[7] Suppl Fig 14A, middle panel and Suppl Fig 18A-C. It is unclear how there can be active plasmin in mouse plasma. Any plasmin generated and released into the plasma would immediately be neutralized by excess levels of inhibitors (alpha2-antiplasmin and alpha2-macroglobulin), unless those inhibitors have been saturated. Under those circumstances, one would expect to see a large increase in plasmin-anti-plasmin complexes, and a "lytic state" in the plasma manifested by bleeding, and degradation of fibrinogen with increased in fibrin degradation products such as d-dimer.

[8] Discussion. Some of the conclusions seem speculative, for example p.10, lines 280-283, and p. 14, lines 382-384.

[9] The authors should at least discuss other fibrinolytic receptors that might contribute to B-ALL. progression. Most important would be uPAR. Do BM stromal cells express uPAR, and to what extent does uPAR contribute to this process? This plasmin-generating pathway would also be impacted by treatment with EACA.

[10] Methods p. 22, line 601. "Plasmin staining" is used throughout the manuscript. Antibodies to plasminogen typically also recognize plasmin. How is it that this staining is specific for plasmin?

[11] Methods p. 26, lines 692-708. Plasmin assay. For some experiments, cells were in 1% FBS, which may contain plasmin inhibitors (eg. alpha2-antiplasmin, alpha2-macroglobulin, and C1 esterase inhibitor). For plasminogen activation, what culture medium was used for the cells, did it contain serum, and were the cells in this medium during the assay?

Minor Concerns:

[1] p. 10, line 285. The wording of this subtitle is a bit awkward. Please revise.

[2] p. 9, line 253: comma is not necessary

[3] p. 15, line 415: "our" findings

Reviewer #2

(Remarks to the Author)

In this paper, the authors used several mouse and human cell lines to elucidate the role of ANXA2 in promoting B-ALL homing and engraftment, focusing their attention on the fibrinolytic pathway. ANXA2 is a well-known molecule with multiple roles, including cell-cell interactions, vesicle fusion, and protease regulation, among others. As the authors recognized in the discussion section, ANXA2 has been the subject of many papers regarding its role in several cancers. However, the mechanisms they described of ANXA2 in BCR-ABL1+ B-ALL are novel, and they also explore the potential clinical relevance of this mechanism with positive results. This study confirmed a well-known interaction between ANXA2 and S100A10, and the need for that interaction for plasminogen activation, using different WT and ANXA2-deficient stromal cell lines. The authors showed with different models that the lack of that interaction prevented the invasion of B-ALL. They used cell lines transplanted into WT and tPA deficient mice and administered plasmin systemically. The authors confirmed that ANXA2 is essential to convert plasminogen to plasmin, as they found decreased levels of plasmin in the bone marrow and plasma of the ANXA2 KO mice. They also found increased levels of IL-6 in WT mice compared to WT mice engrafted with B-ALL, hypothesizing that IL-6 levels are increased because of the effect of TNF-alpha secreted by leukemic cells. Restricted access to growth factors in the BMM leads to reduced mTORC2 activation and proliferation of BCR-ABL1+ cells. ANXA2-deficient MSC reduces the proliferation of BCR-ABL1+ cells and observed reduced pAKT in the leukemic cells and increased co-localization of fibronectin and insulin-like growth factor 1 (IGF-1) in the BMM. The result of a dense ECM that lacks plasminogen can be to trap growth factors such as IGF-1. The authors confirmed that BCR-ABL1+ B-ALL cell numbers increased once IGF1 was released and treating B-ALL cells in a mouse model with an inhibitor of IGF1R increased survival.

Comments to be addressed by the authors:

1. Using a mouse model of ANXA2 KO (KO), the authors demonstrated that there is impaired engraftment of a pre-B BCR-ABL+ cell line in the KO model compared to the wildtype (WT). The authors justify the interest in ANXA2 with supplementary figure 1 and with a submitted publication, where they show that ANXA2 is differentially expressed between osteoblasts from a mouse model engrafted with an AML cell line and CML. The authors provided information comparing osteoblasts from CML and AML, but the manuscript uses KO, mesenchymal stromal cells (MSC), and BCR-ABL+ ALL. The introduction needs to justify the rationale of the model and the experiments. The manuscript relies on a submitted publication, which would be better if it is at least on a pre-print version available to the reader.

2. Supplementary Figure 2 shows no difference in the hematological parameters and number of mature and early progenitors and hematopoietic stem cells between WT and KO models; however, the authors did not assess bone marrow stromal cells to look for any differences in mesenchymal stromal cells and endothelial cells.

3. It is important to highlight that the authors performed the experiment using irradiated and non-irradiated mice. The difference between the KO and WT was more significant when they used non-irradiated mice. The authors should address different explanations for this difference (line 161). As the ANXA2 KO model has ineffective neo-angiogenesis, and irradiation causes an expansion in vessels, it is important to rule out that the decreased engraftment is specific to ANXA2 and not because of a reduced number of vessels in the KO model. The authors should address this possibility. I suggest assessing endothelial and mesenchymal stromal cells by immunohistochemistry and compare the KO and the ANXA2 models before and after engraftment in the non-irradiated and irradiated mice or to use a conditional mouse model where they KO ANXA2 specifically in MSC – osteoblastic cells.

4. The authors were not able to assess the levels of ANXA2 on osteoblasts, following their previous observations, as they claimed that it is difficult to study (Line 178). As an alternative, the authors can use commercially available differentiation media to differentiate mesenchymal stromal cells in vitro into osteoblasts or use another rationale to study MSC.

5. Line 181 refers to non-relevant differences between WT and ANXA2-deficient MSC (from Supplemental figures 10C-D); however, in the figure legend, it does not mention that ANXA2 KO was assessed. However, Supplemental figure 11A does compare the level of expression of ANXA2 between WT and KO, and the KO has lower levels. The author needs to clarify this information.

6. To study the clinical relevance of their findings, the authors used an anti-hemorrhagic drug (EACA) to treat WT and B-ALL engrafted mice. Mice engrafted with BCR-ABL1+ B-ALL cells and treated with EACA showed higher fibronectin levels and a lower leukemic burden, reduced plasmin, plasminogen, and tPA in the bone marrow. Survival was extended after combining EACA and low-dose cytarabine. The authors also demonstrated that the same effect was seen using a human BCR-ABL1+ B-ALL cell line transplanted into an immunodeficient mouse.

7. The authors demonstrated in different experiments that the effect of the fibrinolytic pathways is particular for B-ALL and not for AML or CML. Despite referring to different genetic alterations, they do not offer any explanation of why this phenomenon is exclusively seen in B-ALL. One possible experiment would be to KO BCR-ABL from the B-ALL cell line and explore if it's related to the specific fusion protein in B-ALL.

8. There are not enough details in the methods section or in the results section about the cell lines and the rationale to use them. The authors need to review that all the cell lines are mentioned in methods and provide a rationale to use different ones per experiment. Most of the data presented refer to engraftment; however, proliferation or apoptosis was not measured to provide more information about the mechanism of action of the ANXA2 KO model, to distinguish a defect in homing, proliferation, or induction of cell death.

Reviewer #3

(Remarks to the Author)

In this study, Minciacchi et al. found that activation of plasmin, a key fibrinolytic enzyme, by annexin A2 (Anxa2) distinctly impacts progression of BCR-ABL1+ B-cell acute lymphoblastic leukemia (B-ALL) via modulation of the extracellular matrix (ECM) in the BMM. They suggested that the dense ECM in a BMM with decreased plasmin activity entraps IGF-1 and reduces mTORC2-dependent signaling and proliferation of B-ALL cells. Furthermore, they revealed that treatment with ϵ -aminocaproic acid (EACA), which inhibits plasmin activation, reduced tumor burden and prolongs survival, implicating EACA administration as an adjunct therapy.

Overall, this is an interesting study with a large amount of data, which suggest that fibrinolytic pathway could be targeted to modulate B-ALL progression. However, there are some concerns that need to be addressed in a potential revision.

Major points:

1. In this study, the authors analyzed germline Anxa2-KO mice instead of conditional KO (cKO) mice. Given that Anxa2 is expressed in multiple cell types in the BMM, such as macrophages, fibroblasts, and endothelial cells (Figure S11), one cannot draw definitive conclusion that the phenotypes observed were merely due to MSC-derived Anxa2, unless BMSC-specific cKO mice were analyzed. Better tune down this claim throughout the manuscript.
2. In lines 228-230, the authors indicate that IL-6 may be secreted by B cells. However, in lines 238-241, they speculate that MSC could be the main source of IL-6 in response to TNF α stimulation. Since IL-6 is known to be produced by multiple cell types in the BMM, one should not draw random conclusions on the source of IL-6 without direct evidence (eg. conditional deletion of IL-6 from different cell types). Same problem for the first paragraph of the Discussion section (lines 379-384). Overall, the IL-6 part is very weak.
3. Please avoid citing a manuscript that is under revision (Minciacchi et al.), which is not a formal proof of evidence and might raise potential questions as to conflict of interests between different papers.
4. Incorrect statistical methods were used in many places (e.g Two-way ANOVA should be used for Figure 2C), please double-check throughout the manuscript. There are only two CML and AML samples in Figure S1 with highly variable and non-significant results. It would be better to add more samples to this figure.

Minor points:

1. The authors should specify the differences between inducing CML and B-ALL using BCR-ABL viruses.

2. Line 212: it should be ANXA2-KO, not ANXA2.
3. The error bars for Figures 2F and 3K were blocked by the x-axis.
4. In Figure S10C, representative western-blot image dose not match statistical results. Please show the average image.
5. In Figure S10D, only two mice were analyzed, which cannot draw significant conclusions.

Version 1:

Reviewer comments:

Reviewer #1

(Remarks to the Author)

The authors have addressed the points raised previously. I have no further questions.

Reviewer #2

(Remarks to the Author)

The authors have thoroughly addressed all the comments I raised. They made substantial revisions to the manuscript and conducted new experiments, which satisfactorily address my concerns. I appreciate the time and effort they invested in improving their work.

However, I have one remaining comment regarding the endothelial cell quantification. The images provided for review in the PDF have low resolution, making it difficult to discern the morphology of CD31+ endothelial cells. I recommend using high-quality images in the paper and including a higher magnification image to clearly show endothelial cell morphology. This suggestion arises from the fact that other hematopoietic cells can also stain for CD31. Aside from this, I have no further comments.

Reviewer #3

(Remarks to the Author)

The authors have addressed my concerns. This manuscript is ready to be accepted for publication.

REVIEWER**COMMENTS**

Reviewer #1 (Remarks to the Author); expert in fibrinolysis:

It has been known for some time that the fibrinolytic system, which generates the serine protease plasmin, promotes the release of hematopoietic stem cells from their bone marrow niche. In this interesting study, the authors provide substantial evidence for involvement of the fibrinolytic system, specifically its cell surface arm enabled by the annexin A2 (A2) complex, in remodeling of bone marrow extracellular matrix (BMM) and mobilization of BCR-ABL+ B-cell leukemia cells into the peripheral blood. The authors postulate that the A2 complex generates plasmin which liberates IGF1 from the BMM, allowing mTORC2-mediated signaling and proliferation of ALL cells. It is postulated further that ALL cells increase hepatic synthesis of plasminogen, the precursor of plasmin, creating a cyclical acceleration of the disease process. The authors suggest that adjunctive treatment with a fibrinolysis inhibitor, such as EACA, could reduce tumor burden and prolong survival.

The manuscript addresses a reasonable premise based on previously published literature. For the most part, the experiments appear to have been well-executed, though there are some issues to be resolved. The manuscript contains a large amount of data, much of which resides within the 20 supplemental figures. The work is original, and generally justifies the conclusions. There are, however, a few questions that should be resolved.

Response: We thank the reviewer for calling our study 'interesting' and 'original'.

Major

Concerns:

[1] Figure 1A and others. It is not clear why the authors rely upon fibronectin as a marker of BMM integrity? What is the evidence that fibronectin is a bona fide plasmin substrate in vivo? What about other BMM proteins, and what about fibrin, a primary plasmin substrate?

Response: We thank the reviewer for mentioning these points. First of all, we had shown in previous publications that fibronectin in the bone marrow microenvironment (BMM) is involved in disease aggressivity of an imatinib-resistant form of chronic myeloid leukemia (Kumar et al., Leukemia, 2020) and that fibronectin is an important substrate of matrix metalloproteinase-9 (MMP-9), which controls invasion of B-cell acute lymphoblastic leukemia (B-ALL) in the BMM (Verma et al., Leukemia, 2020). MMP-9 is activated by plasmin (Davis, J Cell Sci, 2001). Secondly, our first observations included increased levels of fibronectin in the BMM of annexin (Anxa)2 KO mice. Based on several publications on the role of ANXA2 for promoting fibrinolysis (Huang, D. et al. Front Med, 2017, and Kwon, M. et al., Front Biosci, 2005) and the involvement of the plasmin system in metastasis and cancer spread (Bazzi Z.A. et al., BMC Cancer, 2016, and He, Y. et al., Clin Cancer Res, 2007) we decided to test plasmin activation in Anxa2 KO mice. Thirdly, we tested levels of laminin and collagen IV in Anxa 2 KO mice and found similar levels of laminin in the BMM of Anxa2 KO mice with and without B-ALL compared to the respective controls (Supplemental Figures 8B-C).

Collagen IV levels were significantly lower in the BMM of Anxa2 KO compared to WT mice (Figure a below).

However, to follow this reviewer's suggestion, we stained bone sections of leukemic (B-ALL) WT and Anxa2 KO mice with an antibody to fibrin or with Masson-Goldner staining. This revealed no differences in fibrin staining in the BMM of leukemic WT and Anxa2 KO mice or WT leukemic mice treated with vehicle versus EACA (Figures b1 and b2 below). It is possible that the higher sensitivity of fibronectin to lysis by plasmin is due to the particular high abundance of fibronectin in the leukemic microenvironment, as has been described for solid tumors (Guerrero-Barberà G et al., Front Cell Dev Biol, 2024) or is due to the fact that fibronectin transcription may be induced by hypoxia (Mao et al., Cell Death Dis., 2023), as found in the BMM (Mendez-Ferrer et al., Nat Rev Cancer, 2020).

Figure a: Immunofluorescence (IF) images for collagen IV (Col IV, green) and 4',6-diamidino-2-phenylindole (DAPI, blue) in bone sections of healthy WT (black) or Anxa2 KO (gray) mice and its quantification (right). The signal intensity is normalized to the number of cells. The scale bar represents 100 μm.

Figure b.1: Immunohistochemistry for fibrin (detected by immunoperoxidase using yellow-brown horseradish-peroxidase chromogen) in bone sections of WT (black) or Anxa2 KO (gray) recipient mice with BCR-ABL1⁺ B-ALL (left) or WT recipient mice with BCR-ABL1⁺ B-ALL treated with vehicle (circles) or EACA (squares) (right).

Figure b.2: Masson-Goldner staining in bone sections of WT (black) or Anxa2 KO (gray) recipient mice with BCR-ABL1⁺ B-ALL (left) or WT recipient mice with BCR-ABL1⁺ B-ALL treated with vehicle (circles) or EACA (squares) (right).

[2] Figure 2I and 2J. Systemic administration of plasmin. If a mouse weighs 30 gm, and the administered dose is 1 mg/kg, each mouse received around 30 ug of plasmin. In a blood volume of 1 ml, this concentration of plasmin (30 ug/ml) would likely be neutralized by its principal inhibitor, alpha 2-antiplasmin (~86 ug/ml) [Lijnen et al. Blood 93:2274, 1999], as well as other inhibitors, such as alpha2-macroglobulin.

Response: We thank the reviewer for pointing this out. We agree in principle. However, as we show in figures 3D-E levels of plasminogen and plasmin are significantly higher in WT mice with B-ALL compared to healthy mice. In contrast, levels of plasminogen activator inhibitor 1 (PAI-1) are significantly lower in WT mice with B-ALL (Figure 3G). Therefore, we believe that calculations about neutralization of plasmin by various inhibitors in the context of B-ALL are unreliable. In addition, having increased the number of samples stained, we now show in Supplementary Figure 13A, that the administered plasmin arrives in the bone marrow and is indeed significantly higher in the BMM of mice treated with plasmin compared to mice treated with vehicle.

However, to address this concern more fully, we have now performed ELISAs for alpha 2-antiplasmin in the plasma of healthy versus leukemic WT or healthy WT versus Anxa2 KO mice, but found no differences (Supplemental figures 14H-I). Plasma levels of alpha 2-macroglobulin are significantly lower in healthy Anxa2 KO compared to WT mice (Supplemental figure 14F), and we find a trend towards lower alpha 2-macroglobulin in WT leukemic compared to healthy mice (Supplemental figure 14G). In conclusion, we believe that the elevated levels of plasminogen and plasmin in B-ALL may prevent the neutralization by alpha 2-antiplasmin or other inhibitors, which themselves may have altered levels in the presence of B-ALL.

[3] Figure 3, panels A, B, D, and E. Plasmin was measured as activity, and concentration estimated by standard curve. Plasminogen was measured by ELISA. To what extent does the plasminogen assay also measure plasmin?

Response: We agree with the reviewer. To measure plasminogen we used a commercially available ELISA kit which employs an antibody against plasminogen. According to the description from the company the kit first enables the binding of plasminogen, plasmin, and plasmin in complex with antiplasmin to the bottom of the plate, and afterwards an antibody is added to detect plasminogen. The text from the product description says: "Mouse plasminogen will bind to the capture antibody coated on the microtiter plate. Plasminogen, plasmin, and plasmin in complex with antiplasmin will react with the antibody on the plate. After appropriate washing steps, polyclonal anti-mouse plasminogen primary antibody binds to the plasminogen." Other kits seem to detect plasminogen in a similar way.

We have communicated with the technical support of the company (abcam), and they agree that the assay cannot fully distinguish between plasminogen and plasmin. Therefore, we have edited the text to reflect this and wrote 'plasmin/-ogen' throughout the text and figures and added 'We cannot exclude that the employed antibody to plasminogen may also be detecting plasmin' to the legends.

[4] Suppl Fig 5E. Secondary survival rates are based on rather low numbers of mice (n=5-6). For the bone marrow result, loss of a single A2KO mouse out of 5 reduces survival by 20%. While this is technically a significant finding (p=0.03), it should be confirmed using larger numbers per group.

Response: While we technically agree with this point, we would like to politely point out that the survival curve in Supplemental Figure 5E is from a secondary transplant from mice with CML. As B-ALL rather

than CML is the focus of the manuscript, we believe that repetition of this secondary transplant will not add value to the manuscript. In the case of the secondary transplant for B-ALL, we believe the differences in death due to non-engraftment in mice receiving B-ALL bone marrow from Anxa2 KO mice, to be so obvious, even with 5-6 mice, that a repetition does not seem reasonable. Given the 3R principles of mouse use we hope the reviewer will agree.

[5] Suppl Fig 5B and 5G. The authors provide evidence that the absence of A2 extends survival in CML, but not AML. What is the presumed mechanism for this based on the biology of the two disorders, and how do these results relate to the subsequent studies focusing on ALL?

Response: We believe that the mechanism for this discrepancy between BCR-ABL1+ diseases (CML and B-ALL) versus MLL-AF9+ acute myeloid leukemia (AML) lies in differential sensitivity to insulin-like growth factor (IGF)1 and differential signaling downstream of the IGF1 receptor. The differences between the two BCR-ABL1+ cell lines K562 (myeloid) and SUPB15 (lymphoid), on the other hand are likely due to differences in expression of IGF1R (Figure 5E) and differences downstream of IGF1R (Figure 5F), which themselves may be due to their different lineage.

We have now dedicated an entire (new) figure (Figure 5) to the mechanism behind the differences between BCR-ABL1+ and MLL-AF9+ leukemias on the one hand, and BCR-ABL1+ myeloid versus lymphoid diseases on the other.

In fact, we show that BCR-ABL1+ BA/F3 cells proliferate more in response to IGF than MLL-AF9+ cells (Figure 4D), and Rictor and pAKT^{S473} levels are higher in BCR-ABL1+ versus MLL-AF9+ cell lines and primary cells (Figures 5A-B). In addition, levels of IGFR are significantly higher on BCR-ABL1+ versus MLL-AF9+ primary leukemia cells (Figure 5D) and BA/F3 cells (Supplemental Figure 16D). In addition, Supplemental Figure 16C shows, that Rictor levels are higher in the BCR-ABL1+ K562, compared to the MLL-AF9+ AML cell line. IGF1 treatment resulted in increased levels of both Rictor and pAKT^{S473} only in NALM6 cells (Figure 5C). Testing differences between the BCR-ABL1+ cell lines K562 and SUPB15, whereby the former is of myeloid and the latter of lymphoid lineage, we found higher expression of IGF1R on SUPB15 cells (Figure 5E). After stimulation with IGF Rictor and pAKT^{S473} were higher in SUPB15 compared to K562 cells (Figure 5F).

Furthermore, the mTOR inhibitor KU-0063794 also significantly decreased the higher number of BCR-ABL1+ BA/F3 cells (Supplementary Figure 15K), suggesting that this pathway is relevant in B-ALL.

Taken together, these data suggest that MLL-AF9+ AML cells rely less on the availability and effects of IGF1 than BCR-ABL1+ cells, and lymphoid BCR-ABL1+ have even higher sensitivity to the effects of IGF1 than the myeloid BCR-ABL1+ cells.

Given the overall good clinical control of CML by tyrosine kinase inhibitors, the great need for better treatments for B-ALL and the fact that, of course, CML and also B-ALL in our murine models (and 30% of adult B-ALL) are BCR-ABL1+, we chose to focus on B-ALL for this study.

[6] Suppl Fig 13A. n= 3-4 for animals receiving vehicle versus plasmin, and the p value seems to be non-significant (p=0.093). Also, what happens when one gives plasmin to WT animals with ALL? Is there accumulation of plasmin and depletion of fibronectin?

Response: We agree with the reviewer. We have now increased the number of analyzed samples. As mentioned in response to point 2 by this reviewer #1, we now show in Supplementary Figure 13A, that the administered plasmin arrives in the bone marrow and is indeed significantly higher in the BMM of mice treated with plasmin compared to mice treated with vehicle. The number of mice per cohort is 9-10, as stated in the legend.

Administration of plasmin to WT mice with B-ALL neither leads to survival prolongation nor a difference in the levels of fibronectin or plasmin in the BMM (Figure c below). We hypothesize that the reason for this lies in the fact that WT mice with B-ALL already have higher levels of plasmin at baseline compared

to *Anxa2* KO mice with B-ALL (Figure 3A). Administration of additional plasmin may, therefore, not have any further effect.

Figure c: (Left) Kaplan-Meier-style survival curve of WT recipient mice with B-ALL treated with PBS (vehicle; solid line) or plasmin (1 mg/kg; dashed line) weekly starting from day 2 after transplant (n=9-10). (Right) Immunohistochemistry for plasmin (detected by immunoperoxidase using yellow-brown horseradish-peroxidase chromogen) in bone sections of WT mice with B-ALL treated with PBS or plasmin as on left starting from day 2 after transplantation, at time of death (n=4). The scale bar represents 50 μ m.

[7] Suppl Fig 14A, middle panel and Suppl Fig 18A-C. It is unclear how there can be active plasmin in mouse plasma. Any plasmin generated and released into the plasma would immediately be neutralized by excess levels of inhibitors (α 2-antiplasmin and α 2-macroglobulin), unless those inhibitors have been saturated. Under those circumstances, one would expect to see a large increase in plasmin-antiplasmin complexes, and a “lytic state” in the plasma manifested by bleeding, and degradation of fibrinogen with increased in fibrin degradation products such as d-dimer.

*Response: We agree with the reviewer. Indeed, we would like to politely point out that the levels of plasmin are extremely low (10^{-3} units/ml). They are even lower for healthy *Anxa2* KO mice or ϵ -aminocaproic acid (EACA)-treated WT mice with B-ALL (Figures 3A, D and Supplemental Figure 19A). As mentioned in our response to point 2 by this reviewer #1, we have now performed ELISAs for α 2-antiplasmin in the plasma of healthy versus leukemic WT or healthy WT versus *Anxa2* KO mice, but found no differences (Supplemental figures 14H-I). Plasma levels of α 2-macroglobulin are significantly lower in healthy *Anxa2* KO compared to WT mice (Supplemental figure 14F), and we find a trend towards lower α 2-macroglobulin in WT leukemic compared to healthy mice (Supplemental figure 14G).*

*However, to address this point we have now performed an ELISA for plasmin-antiplasmin complexes. This revealed no differences in plasmin-antiplasmin complexes between healthy WT and *Anxa2* KO mice, as well as between WT mice with or without B-ALL (Figure d below).*

Figure d: Levels of plasmin-antiplasmin complexes, measured by ELISA in the plasma of healthy WT (black) or Anxa2 KO (gray) (left) mice or WT mice with B-ALL (right) (n=4-5).

We also performed ELISAs to detect D-dimers. Our results show no differences in D-dimers in the plasma of healthy WT versus Anxa2 KO mice (Supplemental Figure 14D), and a significant increase of D-dimers in WT mice with B-ALL compared to healthy WT mice (Supplemental Figure 14E). As mentioned in our response to point 2 by this reviewer #1, we believe that the presence of B-ALL with its possible, indirect induction of hepatic plasminogen generation (Figure 3J-K and Supplemental Figure 14K-L) and possibly other proteins of the pro- and antithrombotic pathways, alters the complex interplay of natural mediators and inhibitors of plasmatic coagulation.

[8] Discussion. Some of the conclusions seem speculative, for example p.10, lines 280-283, and p. 14, lines 382-384.

Response: Thank you for pointing this out. We have made every effort to rewrite the conclusions in a less speculative way and have altered the text accordingly.

[9] The authors should at least discuss other fibrinolytic receptors that might contribute to B-ALL progression. Most important would be uPAR. Do BM stromal cells express uPAR, and to what extent does uPAR contribute to this process? This plasmin-generating pathway would also be impacted by treatment with EACA.

Response: We agree with the reviewer. To address this point, we induced B-ALL in recipient mice deficient for the protease-activated receptor (PAR) 1. This revealed a trend towards reduction of leukemic burden on day 26 after transplantation and significantly prolonged overall survival in PAR1 KO mice with B-ALL (Supplemental Figures 13C-D).

With regards to plasminogen activator, urokinase receptor (uPAR), we demonstrated that uPAR is similarly expressed on WT versus Anxa2 KO mesenchymal stromal cells (MSC) (Figure e below). In addition, levels of uPAR were significantly higher in the plasma of WT mice with B-ALL compared to healthy WT mice (Supplemental Figure 13E).

Figure e: RT-qPCR for plasminogen activator, urokinase receptor (uPAR) *Plaur* in WT (black) and *Anxa2* KO MSC (gray) normalized to GAPDH (n=3).

[10] Methods p. 22, line 601. “Plasmin staining” is used throughout the manuscript. Antibodies to plasminogen typically also recognize plasmin. How is it that this staining is specific for plasmin?

Response: As also mentioned in our response to point 3 by this reviewer #1, we cannot exclude that our antibody to plasminogen also stains plasmin. Therefore, we have rephrased the term ‘plasmin staining’ to ‘plasmin/-ogen staining’. We have added more details on the plasmin/-ogen staining’ to the methods and a word of caution to the result section on page 9, line 259-260.

[11] Methods p. 26, lines 692-708. Plasmin assay. For some experiments, cells were in 1% FBS, which may contain plasmin inhibitors (eg. alpha2-antiplasmin, alpha2-macroglobulin, and C1 esterase inhibitor). For plasminogen activation, what culture medium was used for the cells, did it contain serum, and were the cells in this medium during the assay?

Response: The 1% FBS condition was used only when testing the activation of MMP-9. All the other relevant experiments were performed using incubation buffer (plasmin activation buffer) (10 mM potassium phosphate, 70 mM sodium phosphate, 100 mM lysine buffer, pH 7.5) without serum. This information has been added to the methods.

Minor

Concerns:

[1] p. 10, line 285. The wording of this subtitle is a bit awkward. Please revise.

Response: This subtitle has been altered to ‘Plasmin regulates IGF1 availability by contributing to breakdown of the ECM in an ANXA2-dependent manner.’

[2] p. 9, line 253: comma is not necessary.

Response: This comma has been removed.

[3] p. 15, line 415: “our” findings

Response: ‘Our’ has been altered to ‘these’ findings.

Reviewer #2 (Remarks to the Author); expert in leukaemia and tumour microenvironment:

In this paper, the authors used several mouse and human cell lines to elucidate the role of ANXA2 in promoting B-ALL homing and engraftment, focusing their attention on the fibrinolytic pathway. ANXA2 is a well-known molecule with multiple roles, including cell-cell interactions, vesicle fusion, and protease regulation, among others. As the authors recognized in the discussion section, ANXA2 has been the subject of many papers regarding its role in several cancers. However, the mechanisms they described of ANXA2 in BCR-ABL1+ B-ALL are novel, and they also explore the potential clinical relevance of this mechanism with positive results. This study confirmed a well-known interaction between ANXA2 and S100A10, and the need for that interaction for plasminogen activation, using different WT and ANXA2-deficient stromal cell lines. The authors showed with different models that the lack of that interaction prevented the invasion of B-ALL. They used cell lines transplanted into WT and tPA deficient mice and administered plasmin systemically. The authors confirmed that ANXA2 is essential to convert plasminogen to plasmin, as they found decreased levels of plasmin in the bone marrow and plasma of the ANXA2 KO mice. They also found increased levels of IL-6 in WT mice compared to WT mice engrafted with B-ALL, hypothesizing that IL-6 levels are increased because of the effect of TNF-alpha secreted by leukemic cells. Restricted access to growth factors in the BMM leads to reduced mTORC2 activation and proliferation of BCR-ABL1+ cells. ANXA2-deficient MSC reduces the proliferation of BCR-ABL1+ cells and observed reduced pAKT in the leukemic cells and increased co-localization of fibronectin and insulin-like growth factor 1 (IGF-1) in the BMM. The result of a dense ECM that lacks plasminogen can be to trap growth factors such as IGF-1. The authors confirmed that BCR-ABL1+ B-ALL cell numbers increased once IGF1 was released and treating B-ALL cells in a mouse model with an inhibitor of IGF1R increased survival.

Comments to be addressed by the authors:

1. Using a mouse model of ANXA2 KO (KO), the authors demonstrated that there is impaired engraftment of a pre-B BCR-ABL+ cell line in the KO model compared to the wildtype (WT). The authors justify the interest in ANXA2 with supplementary figure 1 and with a submitted publication, where they show that ANXA2 is differentially expressed between osteoblasts from a mouse model engrafted with an AML cell line and CML. The authors provided information comparing osteoblasts from CML and AML, but the manuscript uses KO, mesenchymal stromal cells (MSC), and BCR-ABL+ ALL. The introduction needs to justify the rationale of the model and the experiments. The manuscript relies on a submitted publication, which would be better if it is at least on a pre-print version available to the reader.

Response: We thank the reviewer for raising these issues. Firstly, we politely point out that in the majority of experiments we are transplanting syngeneic bone marrow transduced with oncogene-expressing retrovirus, not cell lines, into recipient mice. Only figures 8A-B and Supplementary figures 20A, C, D and F rely on the transplantation of human leukemia cell lines or primary human leukemia cells. Secondly, as the reviewer correctly points out, Supplementary figure 1, a volcano plot depicting differential expression of genes in osteoblastic cells (which are of mesenchymal origin) from mice with chronic myeloid versus acute myeloid leukemia in the syngeneic transduction/transplantation models, as well as an accompanying manuscript was the basis for focusing on Anxa 2. This accompanying manuscript has now been accepted for publication in Blood Advances (Minciacchi VR, Blood Adv. 2024 Jul 12: bloodadvances.2024012867. doi: 10.1182/bloodadvances.2024012867).

As stated on page 3, line 84, osteoblastic cells are of mesenchymal origin. They are derived from mesenchymal stromal cells (MSC). However, osteoblastic cells are difficult to isolate and culture, which is why we opted for MSC. MSC, which express higher levels of Anxa2 than osteoblastic cells (Supplemental Figures 9A-C), are used as a model for cells of the bone marrow microenvironment (BMM), but macrophages, endothelial cells and fibroblasts are similarly involved in plasmin activation (Figure 2B, Supplementary Figure 12C). This is now clearly stated on page 7, lines 200-204. Further, we used an

Anxa2 knockout model, as our microarray had shown higher expression of Anxa2 in osteoblastic cells from mice with AML compared to CML. Therefore, in order to understand the function of Anxa2 in leukemia, a knockout model had to be employed. Thirdly, as mentioned on page 6, lines 160-162 we chose to focus on B-ALL as another BCR-ABL1+ disease, as CML is clinically well controlled, better therapies are urgently required for B-ALL and no survival differences were observed between WT and Anxa2 KO mice with AML. In addition, B-ALL can be induced without prior irradiation (Zanetti C et al., Blood, 2021), which allowed us to exclude possible effects of irradiation on the leukemic BMM (Figure 1E).

2. Supplementary Figure 2 shows no difference in the hematological parameters and number of mature and early progenitors and hematopoietic stem cells between WT and KO models; however, the authors did not assess bone marrow stromal cells to look for any differences in mesenchymal stromal cells and endothelial cells.

Response: We agree with the reviewer. We now include data on numbers of endothelial cells, macrophages and MSC, as well as data on MSC differentiation into adipocytes and osteoblastic cells. Our data reveal no significant differences in the percentages of macrophages (Supplemental Figure 11D) or MSC (Supplemental Figure 10C) from WT or Anxa2 KO mice, but endothelial cells are significantly increased in Anxa2 KO mice (Supplemental Figure 11H). There are also no differences between fibroblasts (Supplemental Figure 11E). However, there is a trend towards a lower number of colony-forming units (fibroblast) (CFU-F) when the cells are derived from Anxa2 KO mice (Supplemental Figure 11A). In addition, osteoblastic differentiation is significantly reduced in Anxa2 KO MSC (Supplemental Figure 11B), but adipocytic differentiation is unchanged (Supplemental Figure 11C). There is also no difference in morphology of fibroblasts from WT versus Anxa2 KO mice.

Furthermore, Supplemental Figure 9D shows no differences with regards to the immunophenotype of MSC from WT versus Anxa2 KO mice, and gene expression of these cell types is shown in Supplemental Figures 10A-B.

3. It is important to highlight that the authors performed the experiment using irradiated and non-irradiated mice. The difference between the KO and WT was more significant when they used non-irradiated mice. The authors should address different explanations for this difference (line 161). As the ANXA2 KO model has ineffective neo-angiogenesis, and irradiation causes an expansion in vessels, it is important to rule out that the decreased engraftment is specific to ANXA2 and not because of a reduced number of vessels in the KO model. The authors should address this possibility. I suggest assessing endothelial and mesenchymal stromal cells by immunohistochemistry and compare the KO and the ANXA2 models before and after engraftment in the non-irradiated and irradiated mice or to use a conditional mouse model where they KO ANXA2 specifically in MSC – osteoblastic cells.

Response: We agree and thank the reviewer for raising this point. We have now included immunofluorescence data to test for endothelial cells (using an antibody to CD31) on bone sections of a) healthy WT and Anxa2 KO mice, b) WT and ANXA2 KO mice with B-ALL (irradiated) and c) WT and ANXA2 KO mice with B-ALL (non-irradiated) (Figure f below). These data reveal that there may be reduced CD31 staining in Anxa2 KO compared to WT mice, but there are no obvious differences between irradiated versus non-irradiated Anxa2 KO mice. In response to point 2 by this reviewer #2 we have now also included an enumeration of endothelial cells in WT versus Anxa2 KO mice, which revealed that endothelial cells are significantly increased in Anxa2 KO mice (Supplemental Figure 11H). Although just a quantitative finding, this would argue against the hypothesis that neo-angiogenesis may be impaired in ANXA2 KO mice.

Figure f: Immunofluorescence (IF) images for CD31 (endothelial cells, green) and 4',6-diamidino-2-phenylindole (DAPI, blue) in bone sections of healthy WT or ANXA2 KO mice (top), non irradiated (left) or irradiated (right) WT or ANXA2 KO mice with B-ALL. The scale bar represents 100 μm .

In conclusion, we cannot completely rule out that ANXA2-deficiency-associated lack of angiogenesis (Zhang Y et al., Shock, 2023, and Lin H, Neurosci Lett, 2023) may be contributory to a deceleration of B-ALL-progression. This seems especially relevant, as irradiation would normally increase angiogenesis and this may be prevented by Anxa2 deficiency. This may explain why B-ALL course is even more prolonged in unirradiated Anxa2 KO mice with B-ALL (Figure 1E). However, our immunofluorescence data do not support this, as CD31+ cells do not seem increased in irradiated compared to unirradiated mice with B-ALL. Please note, though, that the bones from these mice were collected when the mice were moribund, i.e. approximately 30 days after transplantation and, therefore, 30 days after irradiation. Unpublished data from our laboratory on ECM proteins other than fibronectin and laminin have shown that irradiation may also affect 'inflammatory' processes in the ECM of the BMM and may make the ECM more 'penetrable' for B-ALL cells. Additionally, non-irradiation may allow better control of the B-ALL progression by the immune microenvironment in the BMM. A paragraph on these possible explanations for the additional survival prolongation in non-irradiated Anxa2 KO mice with B-ALL has been added to the discussion on page 17, lines 456-463. Generating a murine model with MSC- or osteoblastic cell-specific deficiency of Anxa2 would take a year, and an existing model of this type was not available to us.

4. The authors were not able to assess the levels of ANXA2 on osteoblasts, following their previous observations, as they claimed that it is difficult to study (Line 178). As an alternative, the authors can use commercially available differentiation media to differentiate mesenchymal stromal cells in vitro into osteoblasts or use another rationale to study MSC.

Response: We politely point out that we show ANXA2 expression in osteoblastic cells and MSC in Supplemental Figures 1B-D. Anxa2 expression is higher in MSC than in osteoblastic cells (Supplemental Figure 9B). However, in order to validate these results we have now increased the number of replicates and have replaced the original figures in Supplemental Figures 1B-D. These data in Supplemental Figure 1B demonstrate significantly higher levels of Anxa2 in osteoblastic cells compared to MSC, as suggested by our microarray data in Supplementary Figure 1A. In Supplemental Figure 10D we show Anxa2 expression in undifferentiated MSC and MSC differentiated to osteoblasts.

As mentioned in our response to concern 2 by this reviewer #2, we have now added data on the differentiation of MSC to osteoblastic cells and adipocytes. We find a trend towards a lower number of colony-forming units (fibroblast) (CFU-F) when the cells are derived from Anxa2 KO mice (Supplemental Figure 11A). In addition, osteoblastic differentiation is significantly reduced in Anxa2 KO MSC (Supplemental Figure 11B), but adipocytic differentiation is unchanged (Supplemental Figure 11C).

5. Line 181 refers to non-relevant differences between WT and ANXA2-deficient MSC (from Supplemental figures 10C-D); however, in the figure legend, it does not mention that ANXA2 KO was assessed. However, Supplemental figure 11A does compare the level of expression of ANXA2 between WT and KO, and the KO has lower levels. The author needs to clarify this information.

Response: Supplemental Figures 10C-D (in the resubmitted manuscript Supplemental Figures 9A-C) show the relative levels of Anxa2 in primary stromal cells (MSC, osteoblastic cells, endothelial cells) from healthy WT mice by Western Blotting and qPCR. Here, Anxa2 expression was NOT compared between WT and Anxa2 KO mice. In contrast, Supplemental Figures 9D and 10A-B show the immunophenotyping and differential gene expression in MSC from WT versus Anxa2 KO mice, respectively. This section (page 7, lines 187-191) has been reworded to make this more clear.

As the reviewer correctly points out, Supplementary Figure 11A (new Supplemental Figure 11F) compares the expression of ANXA2 in macrophages, MSC and fibroblasts from WT versus Anxa2 KO mice, providing evidence that macrophages and fibroblasts indeed express ANXA2 at similar levels compared to MSC.

6. To study the clinical relevance of their findings, the authors used an anti-hemorrhagic drug (EACA) to treat WT and B-ALL engrafted mice. Mice engrafted with BCR-ABL1+ B-ALL cells and treated with EACA showed higher fibronectin levels and a lower leukemic burden, reduced plasmin, plasminogen, and tPA in the bone marrow. Survival was extended after combining EACA and low-dose cytarabine. The authors also demonstrated that the same effect was seen using a human BCR-ABL1+ B-ALL cell line transplanted into an immunodeficient mouse.

Response: The reviewer is correct, and we agree.

7. The authors demonstrated in different experiments that the effect of the fibrinolytic pathways is particular for B-ALL and not for AML or CML. Despite referring to different genetic alterations, they do not offer any explanation of why this phenomenon is exclusively seen in B-ALL. One possible experiment would be to KO BCR-ABL from the B-ALL cell line and explore if it's related to the specific fusion protein in B-ALL.

Response: We thank the reviewer for bringing up this issue. Please also see our response to concern 5 by reviewer #1. We believe that the mechanism for this discrepancy between BCR-ABL1+ diseases (CML and B-ALL) versus MLL-AF9+ acute myeloid leukemia (AML) lies in differential sensitivity to insulin-like

growth factor (IGF)1 and differential signaling downstream of the IGF1 receptor. The differences between the two BCR-ABL1+ cell lines K562 (myeloid) and SUPB15 (lymphoid), on the other hand are likely due to differences in expression of IGF1R (Figure 5E) and differences downstream of IGF1R (Figure 5F), which themselves may be due to their different lineage.

In the revised manuscript we have dedicated an entire new figure (Figure 5) to this question. Figure 4D in the original and current manuscripts demonstrates that the proliferation of BCR-ABL1+ BA/F3 cells increases in response to insulin-like growth factor (IGF) 1, but this does not apply to MLL-AF9+ BA/F3 cells. Mechanistically, figure 4E (new Figure 5A) reveals that Rictor and pAKTS473 signaling increases in BCR-ABL1+ BA/F3, but not MLL-AF9+ BA/F3 cells after exposure to IGF1. The increased expression of Rictor and pAKTS473 is also shown in primary murine B-ALL cells compared to primary murine AML cells in Figure 4F (new Figure 5B). Furthermore, Figure 4F (new Figure 5D) demonstrates higher expression of IGF 1 receptor (IGF1R) in primary murine B-ALL compared to primary murine AML cells. Additionally, Supplemental Figure 16C uncovers highest expression of mTOR, Rictor and pAKTS473 in the human BCR-ABL1+ K562 cell line and the human (lymphoid) B-ALL NALM6 cell line. NALM6 cells are also the most sensitive to IGF1, as measured by levels of Rictor and pAKTS473, when compared to K562, THP1 and Kasumi cells (Figure 5C). Also in primary murine leukemia cells we show highest expression of IGF1R in BCR-ABL1+ leukemia cells compared to empty vector- or MLL-AF9-transduced cells (Supplemental Figure 16D).

Supplemental Figure 20F demonstrates that treatment with EACA and cytarabine in NSG mice transplanted with THP1 (MLL-AF9+) AML cells does not lead to a survival prolongation and that α 2-antiplasmin is highest in human AML (Supplemental Figure 20H). The area of IGF1 and fibronectin staining in bone marrow sections is positively correlated in B-ALL, but not AML (Supplemental Figures 21E-F). RICTOR is also highest in human B-ALL (Supplemental Figure 21G).

To further address this comment, particularly, with regards to BCR-ABL1+ diseases, we now added a Western Blot testing mTOR, Rictor, p-AKT and AKT in lysates from K562 (BCR-ABL1+) and SUPB15 (BCR-ABL1+) cells. This reveals that Rictor and pAKT^{S473} increase in response to IGF1 only in SUPB15 cells (Figure 5F). We have also added FACS data on IGF1R expression in K562 versus SUPB15 cells revealing that SUPB15 cells express higher levels of IGF1R (Figure 5E). In addition, our xenotransplantation data in Figure 8B uses human B-ALL samples from different patients, all of which are likely associated with various genetic aberrations, but we were not allowed further information on the genetic subtype apart from the information that two samples were BCR-ABL1+. However, despite this presumed genetic variability in the transplanted human B-ALL cells the prolonged survival of recipient mice treated with cytarabine and EACA is uniform. This may suggest that the mechanism is relevant for B-ALL in general and, therefore, may be more relevant in lymphoid than myeloid (BCR-ABL1-negative) leukemia.

Taken together, we believe that a combination of lymphoid phenotype, higher expression of IGF1R and increased signaling via (mTOR), Rictor and pAKTS473 predisposes to increased sensitivity to IGF1 released from the ECM, which is degraded in an Anxa2-dependent manner.

8. There are not enough details in the methods section or in the results section about the cell lines and the rationale to use them. The authors need to review that all the cell lines are mentioned in methods and provide a rationale to use different ones per experiment. Most of the data presented refer to engraftment; however, proliferation or apoptosis was not measured to provide more information about the mechanism of action of the ANXA2 KO model, to distinguish a defect in homing, proliferation, or induction of cell death.

Response: As mentioned in our response to point 1 by this reviewer #2, we politely point out that the great majority of leukemia induction experiments were not performed with leukemia cell lines, but by the

transplantation of syngeneic murine bone marrow that had been transduced with retrovirus expressing the BCR-ABL1 or MLL-AF9 oncoproteins. This information is provided in the materials and methods section under 'Bone marrow transduction/transplantation' on pages 22/23. More detailed information on this murine model is also provided on page 4, lines 126-129. In the 'Cell lines' section in the materials and methods we have now made sure that all cell lines used, as well as the rationale for using them has been added. We have also made every attempt throughout the text to explain why a specific cell line was employed. In brief, we used the murine cell line BA/F3 transduced with empty vector, BCR-ABL1 or MLL-AF9 to mimic the 'normal' condition (control), B-ALL or AML, respectively. We used the human cell line K562 (BCR-ABL1+) as model of CML, THP1 (MLL-AF9+) and Kasumi (AML-ETO+) as models for AML. The MLL-AF9+ THP1 cell line, therefore, is positive for the same oncoprotein as our AML mouse model. NALM6 (BCR-ABL1-negative) and SUPB15 (BCR-ABL1+) cells were used as models of B-ALL.

To answer the reviewer's question about apoptosis we repeated the in vitro assay, in which primary murine B-ALL cells were plated on top of WT or Anxa2 KO MSC embedded in matrigel containing IGF1 and plasminogen. Subsequently, we revealed that the increase of BCR-ABL1+ B-ALL cell numbers in response to IGF1 was due to a decrease of late apoptosis (Supplemental Figure 17).

Reviewer #3 (Remarks to the Author); expert in bone marrow, extracellular matrix:

In this study, Minciacchi et al. found that activation of plasmin, a key fibrinolytic enzyme, by annexin A2 (Anxa2) distinctly impacts progression of BCR-ABL1+ B-cell acute lymphoblastic leukemia (B-ALL) via modulation of the extracellular matrix (ECM) in the BMM. They suggested that the dense ECM in a BMM with decreased plasmin activity entraps IGF-1 and reduces mTORC2-dependent signaling and proliferation of B-ALL cells. Furthermore, they revealed that treatment with e-aminocaproic acid (EACA), which inhibits plasmin activation, reduced tumor burden and prolongs survival, implicating EACA administration as an adjunct therapy.

Overall, this is an interesting study with a large amount of data, which suggest that fibrinolytic pathway could be targeted to modulate B-ALL progression. However, there are some concerns that need to be addressed in a potential revision.

Response: We thank the reviewer for calling our study interesting.

Major points:

1. In this study, the authors analyzed germline Anxa2-KO mice instead of conditional KO (cKO) mice. Given that Anxa2 is expressed in multiple cell types in the BMM, such as macrophages, fibroblasts, and endothelial cells (Figure S11), one cannot draw definitive conclusion that the phenotypes observed were merely due to MSC-derived Anxa2, unless BMSC-specific cKO mice were analyzed. Better tune down this claim throughout the manuscript.

Response: We completely agree with the reviewer. In fact, we show in Figure 2B and Supplemental Figure 12C and state in our response to point 1 by reviewer #2, that Anxa2-deficient macrophages, fibroblasts and endothelial cells show the same decrease of plasmin activation. Following the reviewer's concern, we have toned down the conclusions regarding this point and mention specifically in lines 202-204 and 462-467 that the contribution of other cell types cannot be excluded and are in fact ruled in.

As also mentioned in our response to point 3 by reviewer #2, generating a murine model with MSC- or osteoblastic cell-specific or conditional deficiency of Anxa2 would take a year, and such models were not available to us.

2. In lines 228-230, the authors indicates that IL-6 may be secreted by B cells. However, in lines 238-241, they speculate that MSC could be the main source of IL-6 in response to TNF α stimulation. Since IL-6 is known to be produced by multiple cell types in the BMM, one should not draw random conclusions on the source of IL-6 without direct evidence (eg. conditional deletion of IL-6 from different cell types). Same problem for the first paragraph of the Discussion section (lines 379-384). Overall, the IL-6 part is very weak.

Response: We agree with the reviewer that the reference to IL-6 production by B-ALL cells may be misleading. In actual fact, we have data (Figure g below) showing that B-ALL cells in our model do not make significant amounts of IL-6. Therefore, we have edited this sentence (line 275), including in the discussion (lines 447-449).

We have shown in two previous publications, that B-ALL cells generate significant amounts of tumor necrosis factor (TNF) α (Verma D et al., Leukemia, 2020, and Karantanou C et al., Blood Advances, 2023) (Figure h below). We also showed in the accompanying paper (Minciacchi VR, Blood Adv. 2024 Jul 12:bloodadvances.2024012867. doi: 10.1182/bloodadvances.2024012867) that TNF α increases IL-6 expression in MSC (Figure i below). In the current manuscript, we demonstrated that TNF α (or coculture with SUPB15 cells, which secrete TNF α (Supplemental Figure 14J), increase plasminogen production by the liver (new Supplemental Figure 14K).

To corroborate this experimentally we collected the conditioned medium from human MSC, which had been preexposed to TNF α , which increased their IL-6 production (Minciacchi VR et al., Blood Adv, 2024) (Figure i below). We then cultured Huh7 hepatocellular carcinoma cells in this conditioned medium in the presence or absence of an antibody to IL-6 and measured plasminogen production by the hepatic cells. This revealed a strong trend towards inhibition of the increase in plasminogen in the conditioned medium of Huh7 cells, if the Huh7 cells, cultured in the conditioned medium from the TNF α -exposed MSC, were simultaneously treated with an antibody to IL-6 (Supplemental Figure 14L).

Figure g: Relative expression of IL6 in CD34+ hematopoietic stem cells (HSC), NALM6, SUPB15, THP1 or K562 cells measured by qPCR. The values are normalized to the housekeeping gene GAPDH.

Figure h: Left) TNF α levels in the bone marrow of wildtype mice with CML, AML or B-ALL measured by ELISA. Middle) Relative expression of *TNF* in K562, NALM6 or SUPB15 cells measured by qPCR. The values are normalized to the housekeeping gene GAPDH. Right) TNF α levels in the conditioned medium of K562, NALM6 or SUPB15 cells measured by ELISA (from Karantanou C et al., Blood Adv, 2023).

Figure i: Relative expression of interleukin-6 (IL6) by qRT-PCR analysis at the indicated times in WT (black) versus ANXA5 KO (gray) MSC treated with TNF (15 ng/ml) (from Minciacchi et al., Blood Adv, 2024).

3. Please avoid citing a manuscript that is under revision (Minciacchi et al.), which is not a formal proof of evidence and might raise potential questions as to conflict of interests between different papers.

Response: We thank the reviewer for raising this concern. As also mentioned in our response to reviewer #2, the accompanying manuscript has now been accepted for publication and can be accessed (Minciacchi VR, Blood Adv. 2024 Jul 12;bloodadvances.2024012867. doi: 10.1182/bloodadvances.2024012867).

4. Incorrect statistical methods were used in many places (e.g Two-way ANOVA should be used for Figure 2C), please double-check throughout the manuscript. There are only two CML and AML samples in Figure S1 with highly variable and non-significant results. It would be better to add more samples to this figure.

Response: Thank you for pointing out this error in Figure 2C. This has been corrected and a two-way ANOVA test has been performed. In addition, we have checked all statistical tests employed, performed normality tests on all figures and decided whether parametric or non-parametric tests are required.

We have added more samples to the analyses in Supplementary Figure 1 to n=4 and have replaced the panels accordingly.

Minor

points:

1. The authors should specify the differences between inducing CML and B-ALL using BCR-ABL viruses.

Response: This information can be found in the materials and methods section and on page 4 (lines 126-129).

2. Line 212: it should be ANXA2-KO, not ANXA2.

Response: This has been corrected (line 236).

3. The error bars for Figures 2F and 3K were blocked by the x-axis.

Response: This has been corrected.

4. In Figure S10C, representative western-blot image dose not match statistical results. Please show the average image.

Response: The representative Western blot image in Supplemental Figure 10C (now Supplemental Figure 9A) has been replaced.

5. In Figure S10D, only two mice were analyzed, which cannot draw significant conclusions.

Response: Obtaining enough of these cell types from the BMM to perform these analyses is not trivial. As the reporter mice from Supplemental Figure 10D (now Supplemental Figure 9B) were no longer available to us, we repeated these analyses by sorting MSC, osteoblastic cells and endothelial cells from n= 4 WT mice per cohort using the following immunophenotypes:

- *primary murine osteoblasts: CD4- CD8- Ter119- CD11b- Gr1- CD31- Sca1- CD51+ (Wang W et al. Bio Protoc, 2018)*
- *primary murine MSC: CD45- Ter119- Sca1+ PDFGR α +*
- *primary endothelial cells: CD31+. (obtained by magnetic beads).*

These data were added as Supplementary Figure 9C.

Reviewer #1 (Remarks to the Author):

The authors have addressed the points raised previously. I have no further questions.

Response: We thank the reviewer for this positive response.

Reviewer #2 (Remarks to the Author):

The authors have thoroughly addressed all the comments I raised. They made substantial revisions to the manuscript and conducted new experiments, which satisfactorily address my concerns. I appreciate the time and effort they invested in improving their work.

Response: We thank the reviewer for this positive response.

However, I have one remaining comment regarding the endothelial cell quantification. The images provided for review in the PDF have low resolution, making it difficult to discern the morphology of CD31+ endothelial cells. I recommend using high-quality images in the paper and including a higher magnification image to clearly show endothelial cell morphology. This suggestion arises from the fact that other hematopoietic cells can also stain for CD31. Aside from this, I have no further comments.

Response: We have now included a new panel in Supplemental Figure 11 I which shows the morphology of endothelial cells in lower and higher magnification.

Reviewer #3 (Remarks to the Author):

The authors have addressed my concerns. This manuscript is ready to be accepted for publication.

Response: We thank the reviewer for this positive response.